



Atmospheric
Measurement
Techniques

# Ethane measurement by Picarro CRDS G2201-i in laboratory and field conditions: potential and limitations

**Sara M. Defratyka[1], Jean-Daniel Paris[1], Camille Yver-Kwok[1], Daniel Loeb[1,a], James France[2,3], Jon Helmore[4], Nigel Yarrow[4], Valérie Gros[1], and Philippe Bousquet[1]**

[1]Laboratoire des Sciences du Climat et de l'Environnement (LSCE-IPSL) CEA-CNRS-UVSQ Université Paris Saclay, Gif-sur-Yvette, 91191, France
[2]Department of Earth Science, Royal Holloway University of London, Egham, TW20 0EX, UK
[3]British Antarctic Survey, Natural Environment Research Council, Cambridge, CB3 0ET, UK
[4]National Physical Laboratory (NPL), Hampton Road, Teddington, TW11 0LW, Middlesex, UK
[a]now at: Chemistry Department, Université Paris-Saclay, Orsay 91400, France

**Correspondence:** Sara M. Defratyka (sara.defratyka@lsce.ipsl.fr)

**Abstract.** Atmospheric ethane can be used as a tracer to distinguish methane sources, both at the local and global scale. Currently, ethane can be measured in the field using flasks or in situ analyzers. In our study, we characterized the CRDS Picarro G2201-i instrument, originally designed to measure isotopic $CH_4$ and $CO_2$, for measurements of ethane-to-methane ratio in mobile-measurement scenarios, near sources and under field conditions. We evaluated the limitations and potential of using the CRDS G2201-i to measure the ethane-to-methane ratio, thus extending the instrument application to simultaneously measure two methane source proxies in the field: carbon isotopic ratio and the ethane-to-methane ratio. First, laboratory tests were run to characterize the instrument in stationary conditions. Subsequently, the instrument performance was tested in field conditions as part of a controlled release experiment. Finally, the instrument was tested during mobile measurements focused on gas compressor stations. The results from the field were afterwards compared with the results obtained from instruments specifically designed for ethane measurements. Our study shows the potential of using the CRDS G2201-i instrument in a mobile configuration to determine the ethane-to-methane ratio in methane plumes under measurement conditions with an ethane uncertainty of 50 ppb. Assuming typical ethane-to-methane ratios ranging between 0 and 0.1 ppb ppb$^{-1}$, we conclude that the instrument can accurately estimate the "true" ethane-to-methane ratio within $1\sigma$ uncertainty when $CH_4$ enhancements are at least 1 ppm, as can be found in the vicinity of strongly emitting sites such as natural gas compressor stations and roadside gas pipeline leaks.

## 1 Introduction

Methane ($CH_4$) is the second-most potent anthropogenic greenhouse gas, with an average atmospheric mixing ratio reaching up to 1892 ppb on the global scale in November 2020 (Dlugokencky, 2021), almost 3 times more than during the pre-industrial era. Anthropogenic methane emissions amount to more than half of the total input of methane to the atmosphere and include a range of sources such as landfills, wastewater treatment plants, agriculture, coal, oil and natural gas industries (IPCC, 2013; Turner et al., 2019; Saunois et al., 2020). Large uncertainties remain in the quantification of these sources' magnitudes and locations (Saunois et al., 2016). The variety of methane sources and their geographical overlap increase the difficulty of closing the present methane budget from global to local scale.

Methane sources often co-emit a specific mixture of other gases that can be used as tracers to identify them. For instance, ethane ($C_2H_6$) is associated with thermogenic methane and is therefore co-emitted during extraction of coal, oil and natural gas as well as transportation of the lat-

ter (e.g., Aydin et al., 2011; Hausmann et al., 2016; Helmig et al., 2016; Schwietzke et al., 2014; Sherwood et al., 2017; Simpson et al., 2012). Typically, the $C_2H_6$ mixing ratio in the clean continental atmosphere ranges between 0.5–2 ppb, but it can reach up to 1000 ppb in the vicinity of methane and ethane emitters, such as gas production facilities (Simpson et al., 2012; Rella et al., 2015). In the case of the natural gas industry, a range of values for ethane-to-methane ratio ($C_2H_6 : CH_4$) are observed depending on the geological reservoir from which the gas has been extracted and on its eventual processing. The reported ratios (calculated as molar ratio when based on atmospheric measurements) depend on the type of production facilities and hydrocarbon reservoirs: from 0.01 to 0.06 for gas leaks and gas compressors (Lopez et al., 2017; Lowry et al., 2020; Yacovitch et al., 2014), or higher than 0.3 for processed natural gas liquids (Kort et al., 2016; Yacovitch et al., 2014). Different ratios are also observed in the case of dry gas (0.01–0.06) and wet gas (>0.06). Regarding offshore oil and gas platforms, $C_2H_6 : CH_4$ typically is around 0.05, but ratios of 0.002 and 0.17 have been observed (Yacovitch et al., 2020). Recent studies (Lan et al., 2019; Turner et al., 2019; Yacovitch et al., 2020) showed varying $C_2H_6 : CH_4$ ratios for different facilities, even at the local scale. Lan et al. (2019) showed an increase in $C_2H_6 : CH_4$ at oil and natural gas observation sites in the National Oceanic and Atmospheric Administration Global Greenhouse Gas Reference Network (GGRN) over the course of the respective measurement period. In contrast, biogenic sources such as landfills and cattle farms show either zero or only very small values (<0.002) of $C_2H_6 : CH_4$ ratios (Assan et al., 2017; Yacovitch et al., 2014).

At the local scale, observing changes in $C_2H_6 : CH_4$ provides additional information about specific methane sources, especially in areas with multiple $CH_4$ enhancements from unknown origins (Assan et al., 2017; Lopez et al., 2017; Lowry et al., 2020; Yacovitch et al., 2014, 2020). The currently available techniques, such as gas chromatography with a flame ionization detector (GC-FID) and Fourier-transform infrared spectroscopy (FTIR), provide long-term or short-term (e.g., hourly timescale) measurements of ethane and other components in stationary conditions (Bourtsoukidis et al., 2019; Gros et al., 2011; Hausmann et al., 2016; McKain et al., 2015; Yang et al., 2005; Paris et al., 2021). Laser-based instruments, such as the Los Gatos Research (LGR) Ultraportable Methane/Ethane Analyzer (UMEA; based on a cavity-enhanced absorption technique), the Picarro cavity ring down spectroscopy (CRDS) analyzers (Rella et al., 2015) and the tunable infrared laser direct absorption spectroscopy (TILDAS) analyzer (Smith et al., 2015; Yacovitch et al., 2014), make it possible to measure ethane at high frequency and on a mobile platform. Here, building on previous studies with CRDS instruments, we specify the possibilities and limitations of measuring $C_2H_6$ using the CRDS G2201-i in the vicinity of a methane source. The CRDS G2201-i is originally designed to measure $^{12}CO_2$, $^{13}CO_2$, $^{12}CH_4$, $^{13}CH_4$ and $H_2O$ and records $C_2H_6$ only as an internal way to correct $^{13}CH_4$; thus the observed $C_2H_6$ mixing ratio must be corrected and calibrated.

Previous studies already showed the possibility of using such instruments to determine the $C_2H_6 : CH_4$ ratio in field conditions (Rella et al., 2015; Assan et al., 2017; Lopez et al., 2017, Lowry et al., 2020). In the study of Assan et al. (2017), a CRDS G2201-i was located in a fixed location nearby to natural gas facilities. Over the course of 2 weeks, dried ambient air was measured simultaneously by CRDS G2201-i and GC-FID, using the 10 min averages for 16 "events" of high methane mixing ratios lasting more than 1 h. The $C_2H_6 : CH_4$ ratio allowed the separation of plumes of biogenic or thermogenic origin.

Rella et al. (2015) and Lopez et al. (2017) used the CRDS instrument as part of a mobile setup enhanced with a storage tube, called AirCore (Karion et al., 2010). This storage tube allows sequential reanalysis of air at an improved time resolution and hence precision. The mobile measurements can be made in two modes using this setup. During the "monitoring mode" the air is injected into the analyzer and into the open-ended AirCore at the same time. In the "replay mode", air from the AirCore is measured. Using the AirCore with a lower flow rate increases the sampling frequency. The replay mode is only used after the observation of a methane plume (Rella et al., 2015; Lopez et al., 2017; Hoheisel et al., 2019). Rella et al. (2015) observed $C_2H_6 : CH_4$ ranging from 0.12 for gas sources and 0.22 for oil wells in the Uintah Basin (Utah, USA).

In this study, the main purpose is to evaluate the performances of the CRDS G2201-i and the applicability of making short-term, direct, continuous and mobile measurements of ethane in methane-enriched air, with sufficient precision during near-source surveys. Our motivation is to perform both isotopic and ethane measurements with only one instrument in the field in order to improve the partition of methane sources without the need for an additional analyzer. We aim to provide a protocol useful for other scientific teams that do not possess an analyzer designed for ethane measurements but already have the CRDS G2201-i and intend to use it under field conditions for measuring both $\delta^{13}CH_4$ and ethane-to-methane ratio.

To achieve this goal, the first step consists of laboratory tests to calculate the calibration factors and also to check the instrument performance under laboratory conditions, extending preliminary work by Assan et al. (2017). The second, novel step evaluates the performances of the instrument during mobile field measurements in a controlled release experiment. Therefore, a mixture with known $C_2H_6 : CH_4$ and $CH_4$ emission flux was released and compared to measured ratios from CRDS G2201-i and LGR UMEA. In the third step, the instrument has been evaluated in real field conditions, during car-based surveys conducted at gas compressor stations and one landfill. In this step, measured values were compared to values from gas chromatography and those in

natural gas provided by the operator of the gas compressor stations. These extensive tests allow a full characterization of the CRDS G2201-i instrument for car-based ethane measurements and highlighted the limitations of this instrument when measuring $C_2H_6 : CH_4$.

After presenting material and methods for these three steps (Sect. 2), their results are presented (Sect. 3) and discussed (Sect. 4).

## 2 Material and methods

The CRDS G2201-i (Picarro Inc., Santa Clara USA) used during this study was originally designed for measurement of the mixing ratios of $^{12}C^{16}O_2$, $^{13}C^{16}O_2$, $^{12}C^1H_4$, $^{13}C^1H_4$ and $^1H_2^{16}O$ (hereafter $H_2O$). It operates in three spectral lines: 6057, 6251 and $6029\,\mathrm{cm}^{-1}$. As there is an interference of $^{12}C_2^1H_6$ (hereafter $C_2H_6$) on $^{13}CH_4$ in the absorption spectra, this instrument also measures $C_2H_6$ to correct this interference. Interferences with other species are presented in Appendix A. By default, $C_2H_6$ is not intended for use by standard users. Thus, the measured $C_2H_6$ mixing ratio is neither corrected nor calibrated, and it is stored in private archived files. To use ethane measurements per se, measured $C_2H_6$ values must be first corrected for interferences with $^{12}C^{16}O_2$ (hereafter $CO_2$), $H_2O$ and $^{12}CH_4$. Different interference correction factors are needed in the absence or presence of water vapor (Assan et al., 2017). These correction factors are used and discussed in light of our new tests in Sect. 2.1.1. The water sensitivity test is also described in Sect. 2.1.1.

To ensure comparability and traceability of the ethane measurement, ethane measured by the G2201-i must eventually be linked to a widely used scale. Therefore, ethane values were calibrated before use (Sect. 2.1.2). Finally, $C_2H_6$ values, corrected and calibrated, can be used to determine the $C_2H_6$ correction on $\delta^{13}CH_4$ mixing ratio or, as in this study, to determine the ethane-to-methane ratio. Figure 1 shows the necessary procedure before using $C_2H_6$ measured by CRDS G2201-i.

The same device (CRDS G2201-i CFIDS 2072) was used as by Assan et al. (2017), allowing the determination of possible long-term drift in calibration factors. Additionally, as part of the laboratory tests, continuous measurement repeatability (CMR; used as a precision in Yver Kwok et al., 2015) and Allan variance (Allan, 1966; Yver-Kwok et al., 2015) were determined for the working gases with different $C_2H_6$ mixing ratios (Sect. 2.1.3). Results obtained for CRDS G2201-i are compared with performances of CRDS G2132-i, which also can measure $C_2H_6$ as an additional feature (Rella et al., 2015) and CRDS G2210-i, which is designed for $C_2H_6$ measurements. The characteristic of each instrument is presented in Table 1.

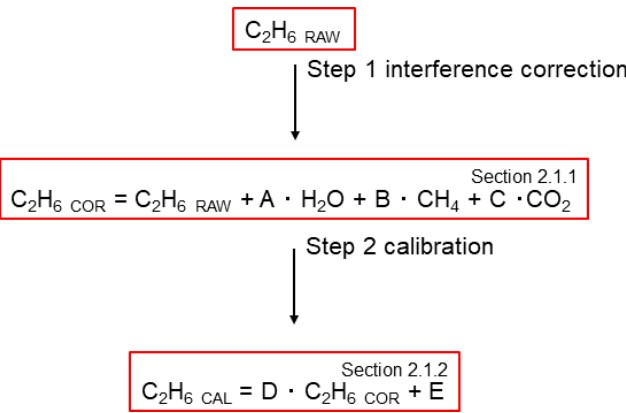

**Figure 1.** Flowchart of steps to use $C_2H_6$ measured by CRDS G2201-i. The number in the corner corresponds to the subsection where methods of each step are presented.

### 2.1 Laboratory setup

### 2.1.1 Sensitivity of interference correction parameters to humidity

The cross-sensitivities with $H_2O$, $CO_2$ and $^{12}CH_4$ induce a bias in raw $C_2H_6$ observed by CRDS G2201-i. Assan et al. (2017) provided the strategy to determine $C_2H_6$ correction factors to account for these interferences. During the experiment, the $C_2H_6$ mixing ratio of measured gas mixture was constant, while the mixing ratio of interfering species was changed and controlled using a setup similar to the one presented in Fig. 2 in Sect. 2.1.2. During one measurement set, the concentration of only one interfering species was changed, while the concentration of other species remained constant. The measurement set was repeated, while varying concentrations of $H_2O$, $CH_4$ and $CO_2$ were adjusted. Using linear regression, the test yielded values for the interference correction factors $A$, $B$, $C$ in Eq. (1):

$$C_2H_{6cor} = C_2H_{6raw} + A \cdot H_2O + B \cdot CH_4 + C \cdot CO_2. \quad (1)$$

The interference of other species in $C_2H_6$ also changes in relation to the water vapor level in the measured sample. In Assan et al. (2017), the correction factors were determined for two different CRDS G2201-i devices (Assan et al., 2017; Table 2). According to that study, if the water vapor level in the measured gas is less than 0.16 % ("low-humidity case"), then interference correction factors are the same for both devices. In the presence of water vapor ($\geq 0.16\,\%$, "high-humidity case"), the correction factors were different for each device. The threshold of 0.16 % corresponds to 26.14 % of relative humidity in standard conditions of temperature and pressure. Due to these differences, drying air is strongly recommended before making measurements (Assan et al., 2017). In this work, the correction factors determined by Assan et al. (2017) are used.

**Table 1.** Characteristics of the instruments used during the study. NaN – data are not available in the data sheet provided by Picarro Inc.

| Analyzer | Species | Rise and fall time | Measurements interval [s] | $CH_4$ operational range [ppm] | $C_2H_6$ operational range [ppm] |
|---|---|---|---|---|---|
| CRDS G2201-i | $CO_2$, $\delta^{13}CO_2$, $CH_4$, $\delta^{13}CH_4$, $H_2O$, $C_2H_6$ (optional) | $\sim 30\,s$ | 3.7 | 1.8–12 | NaN |
| CRDS G2132-i | $CO_2$, $CH_4$, $\delta^{13}CH_4$, $H_2O$, $C_2H_6$ (optional) | $\sim 30\,s$ | 2 | 1.8–12 | NaN |
| CRDS G2210-i | $CO_2$, $CH_4$, $\delta^{13}CH_4$, $H_2O$, $C_2H_6$ | NaN | 1 | 1.5–30 | 0–100 |

**Table 2.** Interference correction on $C_2H_6$ (Assan et al., 2017).

| Humidity | CFIDS 2072 | | | CFIDS 2067 | | |
|---|---|---|---|---|---|---|
| | $A$ [ppm $C_2H_6$/% $H_2O$] | $B$ [ppm $C_2H_6$/ppm $CH_4$] | $C$ [ppm $C_2H_6$/ppm $CO_2$] | $A$ [ppm $C_2H_6$/% $H_2O$] | $B$ [ppm $C_2H_6$/ppm $CH_4$] | $C$ [ppm $C_2H_6$/ppm $CO_2$] |
| Low humidity | $0.44 \pm 0.03$ | $8 \times 10^{-3} \pm 2 \times 10^{-3}$ | $1 \times 10^{-4} \pm 1 \times 10^{-5}$ | $0.44 \pm 0.03$ | $8 \times 10^{-3} \pm 2 \times 10^{-3}$ | $1 \times 10^{-4} \pm 1 \times 10^{-5}$ |
| High humidity | $0.7 \pm 0.03$ | 0 | $3.8 \times 10^{-4} \pm 2 \times 10^{-5}$ | $1 \pm 0.01$ | 0 | $3.9 \times 10^{-4} \pm 2 \cdot 10^{-5}$ |

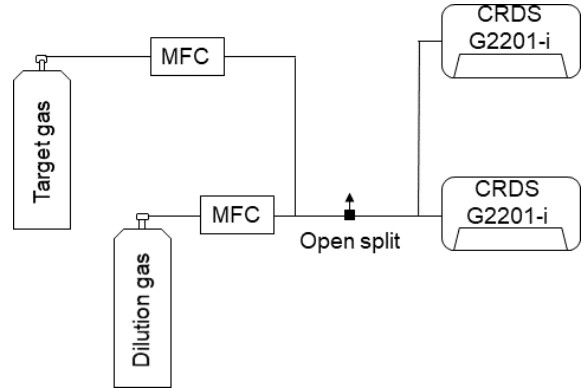

**Figure 2.** Experimental setup used during laboratory tests.

As a part of the laboratory test, we ran a water vapor sensitivity test to revise the parameters of the interference correction (Eq. 1, Table 2) in wet air. The target gas (hereafter referred to as Target Gas 1) had a typical ambient $C_2H_6$ atmospheric mixing ratio. During the test, Target Gas 1 was progressively humidified (0 % to 3 %) by steps of 0.25 %, using a liquid flow controller (Liquiflow, Bronkhorst, Ruurlo, the Netherlands) and a mass flow controller (MFC; Bronkhorst) coupled to a controlled evaporator mixer (CME; Bronkhorst). Each step lasted 20 min, and the cycle was repeated three times. During data analysis, the interference correction factors from Assan et al. (2017) were applied (Table 2). Three cases were tested: no correction, high-humidity case and low-humidity case (except for the first step with dry air, where only the low-humidity correction was applied).

### 2.1.2 Ethane calibration factors

The calibration factors are calculated as the slope (factor $D$) and intercept (factor $E$) of the linear regression of measured (subscripted "cor") $C_2H_6$ versus true $C_2H_6$ ("cal") in Eq. (2).

$$C_2H_{6cal} = D \cdot C_2H_{6cor} + E \qquad (2)$$

Here, the reference gases are prepared using the approach presented by Hoheisel (2018), where a synthetic gas mixture of known $C_2H_6$ ("target") is diluted with a gas ("dilution") with known $CO_2$ and $CH_4$ mixing ratios. The "true" $C_2H_6$ mixing ratio is obtained by applying the following equation:

$$C_2H_{6true} = \left(1 - \frac{1}{2}\left(\frac{CH_{4meas}}{CH_{4dilution}} + \frac{CO_{2meas}}{CO_{2dilution}}\right)\right) \\ \cdot C_2H_{6target2}, \qquad (3)$$

where $C_2H_{6true}$ is the ethane mole fraction in the reference gas obtained by mixing air from the target and dilution cylinders with concentrations of species $X$ (respectively labeled $X_{target}$ and $X_{dilution}$) using MFCs. $CH_{4dilution}$ and $CO_{2dilution}$ are the mixing ratio of the dilution gas. $CH_{4meas}$ and $CO_{2meas}$ are average measured mixing ratios after dilution. This calculation is repeated for different $C_2H_6 : CH_4$ ratios, determined using the MFCs.

The calibration factors are calculated with the $C_2H_6 : CH_4$ ratio gradually increased from 0.00 to 0.15 and measured in steps of 20 min. This measurement cycle is repeated three times. The second target gas has an ethane mixing ratio $\sim 52$ ppm (hereafter referred to as Target Gas 2) and is mixed with the dilution gas via two MFCs (Fig. 2). As the flow rate of the measured gas is greater than the instrument's inlet allowance, an open split is installed before the analyzer to vent

the generated mixture and maintain an ambient pressure at the instrument inlet. The central 15 min of each 20 min measurement is kept for further analysis. Then, the calibration factors are calculated as a regression slope and an intercept of the linear fitting of theoretical (Eq. 3) against measured $C_2H_6$ with already-applied correction factors from Eq. (1). The slope and intercept are used as factors $D$ and $E$ in the calibration equation (Eq. 2). This test was repeated twice: in January 2018 and April 2019.

### 2.1.3 Precision and Allan variance

CMR is calculated as the standard deviation (SD) over different averaging times (see Yver Kwok et al., 2015). The CMR test has been made by measuring a working gas continuously over 24 h. CMR is calculated as the standard deviation over different averaging times (see Yver Kwok et al., 2015). This test was applied twice: first using a working gas with an ambient-air amount of ethane (hereafter referred to as Target Gas 3) and second with a gas mixture where the $C_2H_6 : CH_4$ ratio was equal to 0.05 (mixture of Target Gas 2 and 3). This test is to determine the CMR and instrument noise in the absence or presence of ethane. The Allan deviation was then calculated to determine the noise response of the instrument over different averaging times. Typically, the Allan deviation decreases for increasing averaging time. However, depending on the instrument, with increasing averaging time, instrument drift can lead to the increase in the Allan deviation. Thus, the optimal averaging time can then be identified (Allan, 1966).

Additionally, another target gas (hereafter referred to as Target Gas 4), traceable to the WMO X2004A $CH_4$ scale, was sampled for 20 min, with a $CH_4$ mixing ratio of about 10 000 ppb and a $C_2H_6$ mixing ratio of about 1000 ppb. This test allowed us to determine the linearity and short-term precision of the instrument for a gas with a higher mixing ratio than that of ambient air, both of $C_2H_6$ and $CH_4$.

### 2.1.4 Time drift

The drift of the $C_2H_6$ baseline between December 2018 and May 2019 was investigated. A known working gas (dry atmospheric mixing ratio of $CH_4$ and $C_2H_6$), hereafter referred to as Target Gas 5, was measured during 11 randomly chosen days 20 times over the course of that period and for about 20 min each time. That measurement was made systematically as part of the mobile-measurement protocol (described below). The gas was measured before and after surveys to check instrument stability and the influence of power cycling.

### 2.2 Mobile-measurement setup

This section describes the car-based instrument setup. The general principle of the setup is comparable to previous mobile methane work (e.g., Hoheisel et al., 2019; Lopez et al., 2017; Rella et al., 2015).

As the analyzer was not originally designed for mobile measurements, the vibrations induced by the car motion cause noise in the instrument readouts of $C_2H_6$ mixing ratio. Such a constraint can be overcome using two approaches: firstly, by stopping the car and spending time inside the plume and secondly, by sampling air using the AirCore (Karion et al., 2010; Rella et al., 2015; Lopez et al., 2017) while moving through the plume and eventually reinjecting the AirCore's air into the analyzer while stopped. Previously, the AirCore tool was successfully used as part of a mobile-measurement setup to determine the isotopic composition of the methane source (Rella et al., 2015; Hoheisel et al., 2019; Lopez et al., 2017) and to determine the $C_2H_6 : CH_4$ ratio (Lopez et al., 2017).

Here, both approaches of stopping inside the plume and AirCore replay mode were used during the mobile measurements. The AirCore used in this study is made of a 50 m Dekabon storage tube. In our setup, the instrument flow rate in the monitoring mode was increased to 160 mL min$^{-1}$ (by default, in CRDS G2201-i the flow rate is equal to 25 mL min$^{-1}$) to achieve faster instrument response during mobile measurements. The replay mode was chosen as the optimal solution between increasing the number of measurement points and having enough air for each zone sampled. Here, in the replay mode, using the needle valves, the flow rate decreased by about a factor of 3. With a 50 mL min$^{-1}$ flow rate, one AirCore analysis lasted about 10 min. In the replay mode, the car was stopped to avoid possible increase in instrumental noise due to car vibration. While stopping inside the plume, the data were collected in the monitoring mode with the vehicle engine stopped.

For all mobile measurements, the background mixing ratios for both $CH_4$ and $C_2H_6$ were calculated as the first percentile of the data sampled just before and just after the plumes. Then data with $CH_4$ enhancements above the background are analyzed further. The $C_2H_6 : CH_4$ ratio was calculated for each enhancement as the slope of the linear regression of $C_2H_6$ against $CH_4$. Fitting of the $C_2H_6$ versus $CH_4$ was calculated as a linear regression type II (allowing for uncertainty on both the $x$ and $y$ axis) with the ordinary least squares (OLS) method. Before fitting, both $CH_4$ and $C_2H_6$ were calibrated, and $C_2H_6$ was also corrected (Fig. 1). The measurement setup and data treatment protocol were the same for the controlled release experiment (Sect. 2.3) and for the field experiment (Sect. 2.4).

### 2.3 Controlled release experiment setup

In September 2019, over a period of 5 d, a gas release experiment was conducted by the National Physical Laboratory (NPL; UK) and the Royal Holloway University of London (RHUL; UK). The experiment took place at Bedford Aerodrome, UK. A description of the experimental setup configuration can be found in Gardiner et al. (2017). The goal was to evaluate the methods for calculating $C_2H_6 : CH_4$ ratios, gas

flow rate and isotopic composition during local mobile measurements. Each release lasted about 45 min. During the experiment, the parameters of each release – $C_2H_6 : CH_4$ (0.00 to 0.07), emission flux (up to $70\,L\,min^{-1}$) and the source height (ground level or $\sim 4\,m$ elevation) – were varied. Here, results from 10 releases with known parameters and varying $C_2H_6 : CH_4$ are presented.

Seven releases were measured using the mobile setup (Air-Core and standing in the plume). Air was dried before entering the analyzer using a magnesium perchlorate cartridge. Due to the limited time of the releases, the time spent within the plume was approximately 15 to 20 min. Raw data were corrected according to Eq. (1), using the low-humidity case, and the calibration factors (Eq. 2) were applied.

Three other releases were sampled using $5\,L$ sample bags (FlexFoil, SKC Inc.) only. Between one and three bag samples were collected inside the plume, and one was collected outside the plume as a background. Afterward, bag samples were measured in the laboratory using the CRDS G2201-i. The samples were measured without drying, and the correction was applied for water vapor higher than 0.16 % ("high-humidity case"). Then, the $C_2H_6 : CH_4$ enhancement ratio was calculated separately for each bag and also as a regression slope of $C_2H_6$ against $CH_4$ values. The results are presented in Appendix C.

## 2.4   Field setup and experiment

As a final step to evaluate the G2201-i performance while mobile and under field conditions, the mobile-measurement setup, described in Sect. 2.2, has been used during surveys made in the Paris area. During spring and summer 2019, six surveys focused on three gas compressor stations (one survey for one of them and two surveys for the other two) and one landfill (one survey). All measurements were made outside of the sites from the closest public road. To measure, the car was stationary inside the plumes for about 35 min, and the central 30 min of data was analyzed. Part of the measurements were made with magnesium perchlorate as a dryer before the instrument inlet, and part of the measurements were made without a dryer. This allows for the additional verification of the water influence on the ethane-to-methane ratio observed by the CRDS G2201-i. For each measurement site, three previously evacuated 800 mL flask samples were also taken to be measured within 3 weeks after sampling at the Laboratoire des Sciences du Climat et de l'Environnement (LSCE) (Assan et al., 2017). Measurements were performed with a GC-FID (HP6890) equipped with a CP-$Al_2O_3$ $Na_2SO_4$ column and coupled to a preconcentrator (Entech 2007) to allow automatic injection. A standard cylinder (Messer) containing five non-methane hydrocarbons including ethane was used to check the stability of the instrument, while calibration was performed against a reference standard from the NPL (National Physics Laboratory, UK). A previous characterization of the system had shown that the detection limit covers a few parts per trillion, the reproducibility of measurements is about 2 %, and the precision is better than 5 % (Bonsang and Kanakidou, 2001).

## 3   Results and discussion

### 3.1   Laboratory work

#### 3.1.1   Sensitivity of interference correction parameters to humidity

We estimated the robustness of Eq. (1) interference correction parameters for $H_2O$, $CO_2$ and $CH_4$. Figure 3 shows that without interference correction, the $C_2H_6$ mixing ratio is underestimated, and the instrument displays a negative correlation with water vapor ($r = -0.96$). In the high-humidity-case interference correction, $C_2H_6$ is overestimated and increases with increasing water vapor ($r = 0.86$). Regarding the low-humidity-case interference correction, $C_2H_6$ shows the smallest dependency on water vapor ($r = -0.19$). Applying the low-humidity correction values, the $C_2H_6$ average value is $28 \pm 61$ ppb (standard error 22 ppb), which is similar to the $C_2H_6$ average value obtained during the CMR test ($33 \pm 51$ ppb for raw data) in dry air (Sect. 3.1.3). Overall, according to this study, after applying low-humidity correction values, the water vapor has the smallest impact for observed $C_2H_6$ mixing ratio, and its averaged value is similar to the one obtained in the absence of water vapor. Therefore, the correction factors, determined for the low-humidity case, should also be used in water vapor presence. Our results differ from the findings of Assan et al. (2017), where changing values of the interference correction depending on the humidity were observed. In the absence of further tests to conclude, we recommend drying air for the $C_2H_6$ measurements with the CRDS G2201-i instrument. Details of the water vapor tests are presented in Appendix A.

### 3.1.2   Ethane calibration factors

Here, the calibration slope (factor $D$) and intercept (factor $E$) in Eq. (2) were calculated using linear fitting of true $C_2H_6$ versus observed $C_2H_6$. The calibration factors $D$ and $E$ were determined after applying the interference correction (Eq. 1). Table 3 compares new calibration factors for the specific CRDS G2201-i device CFIDS 2072 obtained in 2018 and 2019 with previous results by Assan et al. (2017). The calibration factors $D$ and $E$ have not changed significantly between 2015 and 2019, indicating that the performance of the instrument remains relatively stable over time.

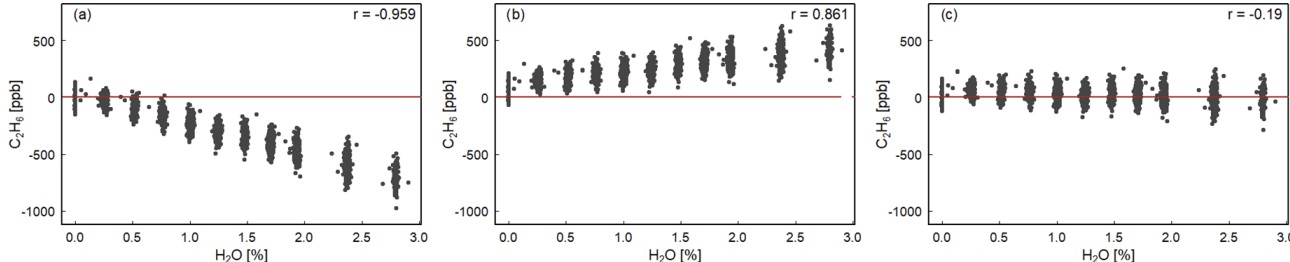

**Figure 3.** $H_2O$ influence on corrected $C_2H_6$. Water vapor is increased in small steps for 4 h while measuring Target Gas 1. The three panels show the result of applying different water correction protocols for the next steps: **(a)** no correction, **(b)** high-humidity interference correction, **(c)** low-humidity interference correction. In all cases, for $H_2O = 0.00\%$, $C_2H_6$ is corrected using low-humidity interference correction. The red line represents 0 ppb.

**Table 3.** Summary of the calibration factors for the CRDS G2201-i device CFIDS 2072.

| $C_2H_6$ calibration | Slope factor $D$ | Intercept [ppm] factor $E$ | Reference |
|---|---|---|---|
| February 2015 | $0.49 \pm 0.03$ | $0.00 \pm 0.01$ | Assan et al. (2017) |
| October 2015 | $0.51 \pm 0.01$ | $-0.06 \pm 0.04$ | Assan et al. (2017) |
| January 2018 | $0.51 \pm 0.01$ | $-0.03 \pm 0.01$ | This study |
| April 2019 | $0.54 \pm 0.01$ | $-0.03 \pm 0.01$ | This study |

### 3.1.3 Precision and Allan variance

We determined the instrument CMR and Allan variance by measuring Target Gas 3 for 24 h. The same gas was also measured by GC-FID coupled to a preconcentrator, yielding a $C_2H_6$ mixing ratio of $2.2 \pm 0.1$ ppb. Using the CRDS G2201-i, the corrected and calibrated value is different and steadily equals $33.2 \pm 1.7$ ppb over the 24 h duration. This value suggests a bias of 31 ppb at low $C_2H_6$ concentrations, which is on the level observed for the ambient air. This bias probably comes from the fact that Target Gas 2 concentration is not known with a good enough precision, leading to errors when diluting to very low concentrations. To remove this bias, $C_2H_6$ mixing ratios were taken as enhancements over background during mobile measurements (Sect. 3.2 and 3.3). For more demanding purposes, a calibration strategy with more measurement points in the lower $C_2H_6$ concentration range and calibration tanks with lower uncertainty should be used.

Following the 24 h test, CMR and Allan deviation (Fig. 4) are calculated for target gases with different $C_2H_6$ mixing ratios: low mixing ratio (Target Gas 3), 100 ppb (mixture of Target Gas 2 and 3) and 1000 ppb (Target Gas 4). In all cases, increasing the ethane mixing ratio does not affect the determined CMR and Allan deviation. Looking at the raw data (one data point every 3.7 s) for different mixing ratios, CMR and Allan deviations are $\sim 50$ and 25 ppb, respectively. Increasing averaging time improves these parameters, and for 1 min average, both CMR and Allan deviations achieve $\sim 13$ ppb. For the CRDS model G2132-i, also not originally designed for ethane measurements (Rella et al., 2015), the CMR in 1 min is $\sim 20$ ppb, and Allan deviation in 1 min is $\sim 25$ ppb. Currently, new CRDS instruments designed for ethane measurements are available, for example, the CRDS 2210-i, which also measures $\delta^{13}CH_4$. Recently (in February 2020), at the Integrated Carbon Observation System (ICOS) Atmosphere Thematic Centre (ATC) Metrology Laboratory (MLab), the CRDS G2210-i was tested, and for $C_2H_6$ its CMR and Allan deviations are equal to 0.9 and 0.8 ppb in 1 min (ATC MLab, personal communication), which is much lower in comparison to our analyzer. However, as stated before, our motivation is to evaluate if any G2201-i (including former ones still operating in many places) can provide scientifically useful ethane measurements. The comparison between the instruments is presented in Table 4.

With a possible 30 ppb bias and a CMR of 50 ppb, the CRDS G2201-i cannot be used to measure an absolute value of ethane in ambient air. However, this instrument can be used to observe ethane enhancement near the source and to estimate $C_2H_6 : CH_4$ ratios. From these numbers, we can deduce that the smallest enhancement that the analyzer can measure with significant precision at the highest possible data acquisition frequency is above 50 ppb. This value was obtained for gas with both a low and high $C_2H_6$ mixing ratio ($\sim 100$ ppb and $\sim 1$ ppm). One can assume that a $C_2H_6$ enhancement is significant when the maximum $C_2H_6$ mixing ratio at the peak is higher than 2 times the CMR, i.e., 100 ppb above background.

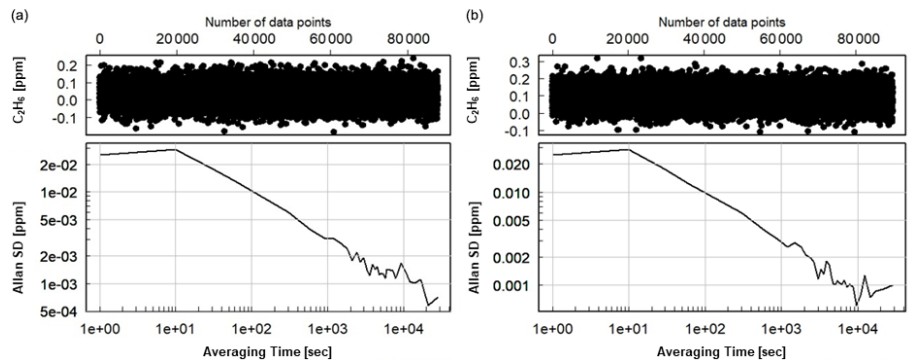

**Figure 4.** Allan deviation for corrected and calibrated $C_2H_6$. **(a)** Measurement of working gas with ambient $C_2H_6$ mixing ratio (Target Gas 3), **(b)** measurement of the mixture of working gas with $\sim 100$ ppb of $C_2H_6$ (mixture of Target Gas 2 and 3).

**Table 4.** CMR and Allan deviation for G2201-i G2132-1 and G2210-i. NA – no data from the test are available.

| Averaging time | ID | G2201-i Low $C_2H_6$ | G2201-i $\sim 100$ ppb $C_2H_6$ | G2201-i $\sim 1000$ ppb $C_2H_6$ | G2132-i (Rella et al., 2015) | G2210-i (ATC MLab) (personal communication) |
|---|---|---|---|---|---|---|
| Raw data | CMR [ppb] | 51 | 50 | 50 | NA | 4.6 |
| | Allan deviation [ppb] | 25 | 25 | 26 | NA | NA |
| 10 s | CMR [ppb] | 30 | 29 | 30 | NA | NA |
| | Allan deviation [ppb] | 29 | 29 | | NA | NA |
| 1 min | CMR [ppb] | 13 | 12 | 12 | 20 | 0.9 |
| | Allan deviation [ppb] | 13 | 12 | 12 | 25 | 0.8 |

### 3.1.4 Time drift

Figure 5 shows the time series of Target Gas 5 measurements with an ambient amount of $C_2H_6$ during the period of December 2018–May 2019. The $C_2H_6$ mixing ratio measurements do not change significantly here. Their mean is equal to $23 \pm 12$ ppb (Fig. 5). It is in contrast to Assan et al. (2017), where a time drift of the baseline was observed. This difference can be caused by the fact that during previous studies, the drift was determined for corrected but uncalibrated data. Here, we applied both correction and calibration before determination of time drift. Moreover, during studies of Assan et al. (2017), bigger changes in determined calibration factors were observed over time (i.e., 60 ppb difference of factor $E$). Our tests showed that the ethane measurements are stable over annual timescales once proper interference correction and calibration are applied. Again, measuring dry air is recommended (Sect. 3.1.1). In the following analyses, no baseline drift correction is applied.

It should be noted that the $C_2H_6$ concentration of Target Gas 5 was in the range of clean continental air (0.5–2 ppb). The observed mean $C_2H_6$ mixing ratio for Target Gas 5, equal to 23 ppb, is overestimated. This is comparable to the 31 ppb bias observed during 24 h measurements of Target Gas 3 (Sect. 3.1.3).

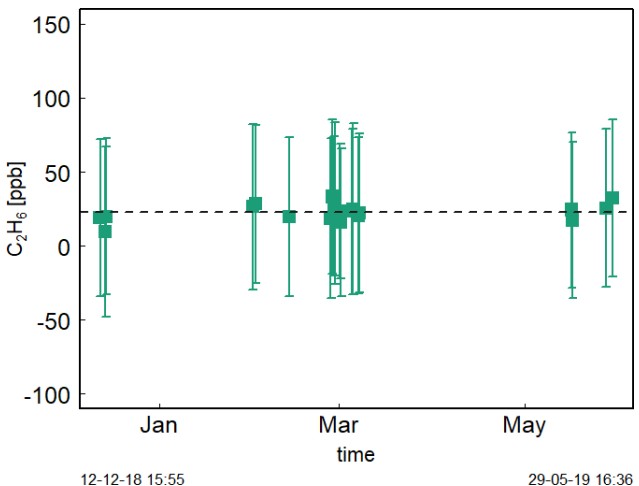

**Figure 5.** Measurements (20 min) of Target Gas 5 over half a year. For each measurement point, squares represent averaged value and error bars $-1$ standard deviation.

### 3.2 Controlled release experiment

Figure 6 and Table 5 show $C_2H_6 : CH_4$ ratios, expressed in ppb ppb$^{-1}$, measured in situ during the controlled release experiment (see Sect. 2.2). During these seven releases, the $C_2H_6 : CH_4$ ratio was set to $\sim 0.032$ for one release, $\sim 0.00$ for two releases and $\sim 0.07$ for four releases. In the case

when $C_2H_6 : CH_4 = 0.00$, ethane was not detected and therefore not released, while methane was released. Possibly, the observed ethane mixing ratio could be due to ethane impurity in the released methane (however, no ethane was detected using the LGR instrument during the zero-ethane releases). For measurements during which the car stopped inside the plume, most of the data from the CRDS G2201-i were lower than the known emitted $C_2H_6 : CH_4$ ratio (mean absolute deviation = 0.011, standard deviation = 0.004), with residuals in the range of $-0.018$ to $-0.002$ for raw data (Table 5). The residuals were calculated as a difference between measured and released $C_2H_6 : CH_4$. The observed underestimation can be caused by a systematic bias observed during laboratory testing or an insufficient number of measurement points (15–20 min of measurement). For AirCore measurements, there are more discrepancies than for the stationary in-plume situation, with residuals in the range of $-0.025$ to $0.027$ (mean absolute deviation = 0.017, standard deviation = 0.009). Thus, the stationary in-plume situation setup shows data with less spread than AirCore results. These results show that in the case of $C_2H_6 : CH_4$ measurements, standing inside the plume gives results which are closer to the reality than AirCore sampling. The example of observed $CH_4$ and $C_2H_6$ mixing ratios while standing inside the plume during one of the gas releases is presented in Appendix B.

We also investigated the sensitivity of the $C_2H_6 : CH_4$ ratio to emission rates. During releases there were two different emission rates: 38 and about $70\,L\,min^{-1}$. For the higher emission rate, the measurements and results were combined when the emission rates were 70, 72 and $73\,L\,min^{-1}$. The $C_2H_6 : CH_4$ ratio was better estimated by the measurements with higher emission rates (bias is divided by more than 2 when increasing flow rate from $\sim 38$ to $\sim 70\,L\,min^{-1}$). This applies to both stationary measurements and using the Air-Core sampler. However, only two different emission rates were implemented, and most of the releases occurred at the rate of $70\,L\,min^{-1}$, limiting the representativity of this sensitivity.

In Table 5 we also report residuals of $C_2H_6 : CH_4$ independently measured by RHUL using an LGR UMEA in an additional car. The residuals in $C_2H_6 : CH_4$ ratios of LGR UMEA are in the range of $[-0.015, -0.001]$, and the mean value is $-0.0051$ (mean absolute deviation = 0.0051). Therefore, the LGR UMEA is predictably more accurate than the CRDS G2201-i standing inside the plumes (CRDS residuals in the range of $-0.018$ to $-0.002$, with a mean of $-0.011$). Despite the observed differences, results obtained by these two methods are comparable, and both instruments were capable of resolving the variation in the $C_2H_6 : CH_4$ ratio in this release experiment.

During the controlled release experiment, we showed that the CRDS is able to separate the different emitted mixtures through their $C_2H_6 : CH_4$. Standing in the plume resulted in a better agreement with the real ratios, with less spread of the residuals than using AirCore sampling. Increasing the Air-

Core sampling frequency could potentially help resolve this discrepancy.

## 3.3 Fieldwork

Measurements were collected in the Paris area downwind of three gas compressor stations (referred to as Ga, Gb, Gc) and one landfill (L). All measurements in this section were done stationary inside the plume.

Table 6 presents values based on raw data (i.e., at $\sim 3.7\,s$ acquisition frequency). We postulate that mobile applications usually aim at the highest possible acquisition frequencies. However, as the 10 s averaging increases $r^2$ fitting by about a factor of 2, comparison of raw data and 10 s averaged data is presented in Appendix D. $C_2H_6$ and $CH_4$ mixing ratios are taken as enhancements over background ($\Delta$). Slopes are calculated using a linear regression type II (uncertainty on the $x$ and $y$ axis influences fitting) with the ordinary least squares (OLS) method. The data are not weighted. Uncertainties reported in Tables 6 and 7 are linearly fitting slope uncertainties without adding uncertainties in $C_2H_6$ measurements.

Campaigns Ga1, Gb1 and Gb2 (Table 6) were made without using a dryer before the instrument inlet. Due to previous results that have cast doubts about the water vapor correction, the high humidity measurements have been rejected from further analysis. Also, in the case of measuring wet air, the ethane-to-methane ratio was significantly higher than expected values provided by the operator. Surveys Gb2 and Gc1 exhibited the highest uncertainties in the estimated ratio and the lowest correlation between the two species. These two surveys had the lowest $CH_4$ enhancements above background, about 0.5 ppm. Based on error propagation (Taylor, 1997) and using 2 times the CMR (100 ppb) as a $C_2H_6$ detection threshold, for a typical $C_2H_6 : CH_4$ of interest of about 0.1, the minimal $CH_4$ enhancement above background would therefore be equal to 1 ppm. It suggests that a minimum $CH_4$ enhancement of 1 ppm could be required to calculate ethane-to-methane ratio in field conditions with this instrument. As our observations are in line with the error propagation, we use 1 ppm $CH_4$ enhancement above background as a detection limit to use the CRDS G2201-i to determine ethane-to-methane ratio in the field conditions close to the methane source and exclude Gb2 and Gc1 from subsequent analysis.

Figure 7 presents observations from the valid cases. We compared the observed ratios with the values provided by the owner of the gas compressor stations. The comparison is presented in Table 7. The residuals between values measured by CRDS and values provided by the owner (considered to be the "true" values) are in the range of $-0.006$ to $0.009$. This range is more symmetrically distributed around the released value than for the controlled release experiment ($-0.018$ to $0.002$; Sect. 3.2). The uncertainty in $C_2H_6 : CH_4$ measured using the CRDS G2201-i in field conditions is smaller than the differences between the ratios of $CH_4$ sources (e.g., biogenic sources $C_2H_6 : CH_4 \sim 0.00$, natural gas leaks and

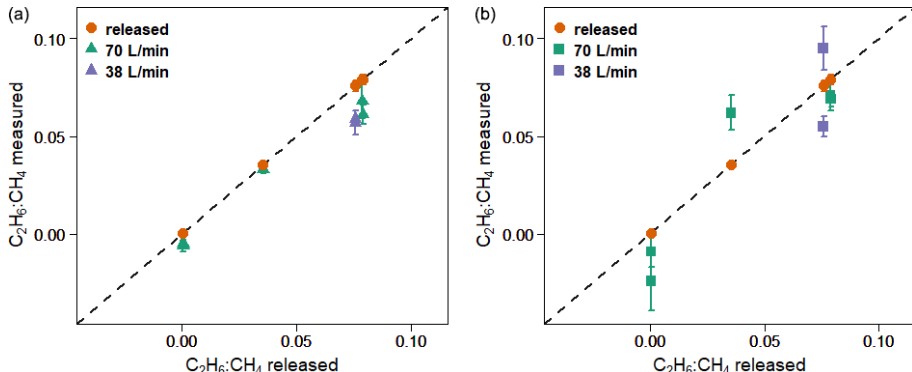

**Figure 6.** $C_2H_6 : CH_4$ observed using G2201-i as a part of a mobile setup. **(a)** Measured standing inside the plumes. **(b)** Measured using AirCore. Red points: known released $C_2H_6 : CH_4$. Error bars represent 1 standard deviation. The uncertainties in released values are invisible on the graph. CE1 💬

**Table 5.** $C_2H_6 : CH_4$ with residuals for non-averaged data observed using G2201-i as a part of a mobile setup while standing inside the plume or from AirCore (AC) measurements. Background subtracted for both $C_2H_6$ and $CH_4$ before determination of $C_2H_6 : CH_4$.

| Emitted $C_2H_6 : CH_4$ | Emitted emission flux [L min$^{-1}$] | Source height [m] | $n$ | LSCE CRDS G2201-i | | | | RHUL LGR UMEA |
|---|---|---|---|---|---|---|---|---|
| | | | | $C_2H_6 : CH_4$ | Residuals | $C_2H_6 : CH_4$ AC | AC residuals | Residuals $C_2H_6 : CH_4$ |
| $0.0355 \pm 0.0011$ | 70 | 4 | 382 | $0.033 \pm 0.002$ | $-0.002$ | $0.034 \pm 0.002$ | $0.027$ | $-0.004$ |
| $0.0788 \pm 0.0025$ | 72 | 4 | 149 | $0.068 \pm 0.009$ | $-0.011$ | $0.070 \pm 0.010$ | $-0.008$ | $-0.006$ |
| $0.0790 \pm 0.0025$ | 73 | 0 | 220 | $0.061 \pm 0.005$ | $-0.018$ | $0.063 \pm 0.006$ | $-0.010$ | $-0.001$ |
| $0.0758 \pm 0.0028$ | 38 | 0 | 142 | $0.059 \pm 0.004$ | $-0.017$ | $0.058 \pm 0.004$ | $-0.020$ | $-0.007$ |
| $0.0758 \pm 0.0028$ | 38 | 4 | 191 | $0.057 \pm 0.006$ | $-0.018$ | $0.057 \pm 0.006$ | $0.019$ | $-0.015$ |
| $0.0005 \pm 0.0006$* | 70 | 0 | 350 | $-0.005 \pm 0.001$ | $-0.005$ | $-0.005 \pm 0.002$ | $-0.025$ | $-0.004$ |
| $0.0005 \pm 0.0006$* | 70 | 4 | 202 | $-0.006 \pm 0.003$ | $-0.007$ | $-0.005 \pm 0.004$ | $-0.010$ | $-0.001$ |
| Mean residuals | | | | | $-0.011$ | | $-0.004$ | $-0.0051$ |

* Small amount of ethane impurity in the methane.

compressor stations $\sim 0.06$, processed natural gas liquids $\sim 0.30$). These results clearly show that $C_2H_6 : CH_4$ measured by the CRDS G2201-i can be used to partially infer the origin of the $CH_4$ during mobile measurements.

Finally, $C_2H_6$ mixing ratios measured by the CRDS G2201-i are compared with results from GC-FID. Three flask samples were taken from every surveyed site and measured afterward in the laboratory using GC-FID. Then, the average of these three measures was calculated, and for all sites their standard deviation is smaller than 1 ppb. In Fig. 8, flask results are compared to results obtained by the CRDS G2201-i during the time of flask sampling. It should be kept in mind that due to the very short sampling time ($<3$ s), the comparison of concentrations is only indicative. For the landfill, the $C_2H_6$ mixing ratio measured by GC-FID is 4.9 ppb, which is higher than typical $C_2H_6$ mixing ratio observed for a clean atmosphere (0.5–2 ppb). For Ga and Gc gas compressor stations, the $C_2H_6$ mixing ratio, measured by GC-FID, is 20.5 and 13.7 ppb, respectively. After subtracting the determined bias, for the landfill and two compressor stations (Ga and Gc), the $C_2H_6$ mixing ratio measured by CRDS is still higher than the one measured by GC- FID (Fig. 8) but within the in-

strument noise. A different situation is observed in the case of the gas compressor station Gb, where a higher $C_2H_6$ mixing ratio is observed. The results from flask samples are higher by about 24 ppb than from the CRDS analyzer after subtraction of 31 ppb bias, which is still within the instrument noise. For all sites, the CRDS measurements show a standard deviation that is almost equal to the averaged value over the sampling time. It is caused by the high instrument noise ($\sim 50$ ppb CMR and 25 ppb Allan deviation for raw data) and short sampling time (less than 1 min).

Fieldwork allowed us to compare our measurements with the operator values and GC measurements. This confirms that this instrument can distinguish between sources and that it agrees within its uncertainty with more precise methods such as GC.

## 4   Synthesis and discussion: overall comparison with other instruments and methods

We determined that using the CRDS G2201-i in a mobile setup to measure $C_2H_6 : CH_4$ in methane plumes is possi-

**Table 6.** Ratio measured at three different gas compressor stations (Ga, Gb, Gc) and a landfill (L); $\Delta CH_4$ and $\Delta C_2H_6$ are defined as the difference between background value (first percentile) and the observed value inside the peak.

| ID | Max $\Delta CH_4$ [ppm] | Max $\Delta C_2H_6$ [ppm] | $C_2H_6 : CH_4$ 1 s | $r^2$ fitting | $n$ (data point) | Date (dd.mm.yyyy) |
|---|---|---|---|---|---|---|
| Ga2 | 1.737 | 0.269 | $0.060 \pm 0.005$ | 0.195 | 533 | 16.05.2019 |
| Ga3 | 5.85 | 0.414 | $0.045 \pm 0.002$ | 0.489 | 495 | 15.07.2019 |
| Gb3 | 1.454 | 0.260 | $0.052 \pm 0.007$ | 0.082 | 613 | 12.07.2019 |
| Gb4 | 1.677 | 0.236 | $0.046 \pm 0.008$ | 0.086 | 336 | 12.07.2019 |
| L | 1.516 | 0.266 | $0 \pm 0.006$ | 0 | 712 | 16.05.2019 |
| Ga1[a] | 1.486 | 0.309 | $0.070 \pm 0.013$ | 0.162 | 138 | 16.05.2019 |
| Gb1[a] | 7.314 | 0.878 | $0.090 \pm 0.001$ | 0.852 | 811 | 27.05.2019 |
| Gb2[a] | 0.513 | 0.323 | $0.085 \pm 0.022$ | 0.024 | 594 | 12.07.2019 |
| Gc1[b] | 0.495 | 0.284 | $0.091 \pm 0.037$ | 0.037 | 711 | 28.05.2019 |

Numbers after identification letters refer to different surveys. [a] Ga1, Gb1 and Gb2 (wet air) and [b] Gc1 (low enhancement) are rejected from further analysis (see text).

**Table 7.** Comparison of results obtained by CRDS G2201-i with the values from the operator company. NA – no data are available.

| ID | CRDS 1 s $C_2H_6 : CH_4$ | Operator data $C_2H_6 : CH_4$ | Residuals $C_2H_6 : CH_4$ | Date (dd.mm.yyyy) |
|---|---|---|---|---|
| Ga2 | $0.060 \pm 0.005$ | 0.051 | 0.009 | 16.05.2019 |
| Ga3 | $0.045 \pm 0.002$ | 0.049 | $-0.004$ | 15.07.2019 |
| Gb3 | $0.052 \pm 0.007$ | 0.052 | 0.000 | 12.07.2019 |
| Gb4 | $0.046 \pm 0.008$ | 0.052 | $-0.006$ | 12.07.2019 |
| L | $0 \pm 0.006$ | NA | NA | 16.05.2019 |

ble and can provide useful scientific results under specific conditions. In laboratory conditions, during measurements of gas containing $C_2H_6$, the CRDS G2201-i has a better CMR (12 ppb in 1 min) and a smaller noise calculated from Allan deviation ($\sim 10$ ppb in 1 min) than those in the CRDS G2132-i, another isotopic analyzer, which are equal to 20 and 25 ppb, respectively, in a 1 min timeframe (Rella et al., 2015). However, both instruments have lower performance than the CRDS G2210-i, designed to measure $C_2H_6$. For the latter instrument, both CMR and Allan deviation are smaller than 1 ppb (ATC MLab test, personal communication). Additionally, based on a literature comparison, for both CRDS instruments, CMR and noise are higher than those obtained from the instrument based on the TILDAS method, designed for mobile measurements of $C_2H_6$ (as described by Yacovitch et al., 2014). For that instrument, the CMR is as low as 19 ppt in stationary conditions and 210 ppt in motion.

The correction of the sensitivity to other species is necessary (Eq. 1) to account for the different instrument responses to a water level lower or higher than 0.16 % (low and high humidity). In this study, during laboratory work, the water vapor sensitivity was evaluated, and the results showed that applying interference correction factors determined for low humidity gave better results, including wet air measurements. This is in contrast to the results obtained by Assan et al. (2017). Rella et al. (2015) noted that less than 0.1 % of the

measured air should be water vapor. Therefore, we consider that water vapor should be removed if at all possible, and we recommend drying air before $C_2H_6$ measurements using CRDS G2201-i.

Previously, the CRDS G2201-i device CFIDS 2072 has only been used in stationary fieldwork over 2 weeks (Assan et al., 2017) to make continuous measurements of $CH_4$, $\delta^{13}CH_4$ and $C_2H_6$ from gas facilities. The CRDS G2201-i and GC-FID measured air simultaneously from the shared inlet and were located 200–400 m from the gas facilities (pipelines and compressors). The GC-FID used in Assan et al. (2017) was a field instrument described in Gros et al. (2011) and Panopoulou et al. (2018) which has an overall uncertainty estimated to be better than 15 %. To obtain identical timestamps as the GC-FID, corrected and calibrated CRDS data were averaged for 10 min every 30 min. Moreover, during that study, flask samples were collected and further analyzed in the laboratory. $C_2H_6 : CH_4$ from flask samples allowed the distinction of methane emissions from the two pipelines. The natural gas in pipeline 1 had an ethane-to-methane ratio equal to $0.074 \pm 0.001$ and for pipeline 2 equal to $0.046 \pm 0.003$. These values are in good agreement with on-site GC-FID results, which reached 0.075 and $0.048 \pm 0.003$ for pipeline 1 and 2, respectively (Assan et al., 2017). Thus, the laboratory values showed good agreement

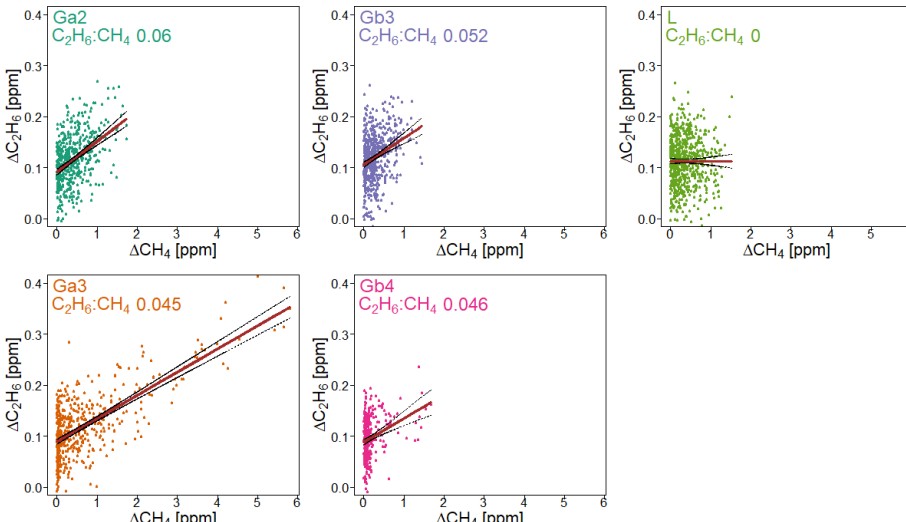

**Figure 7.** $C_2H_6 : CH_4$ for gas compressor stations (Ga and Gb) and the landfill (L), calculated for non-averaged data. Linear fitting (red line) with confidence intervals (black lines).

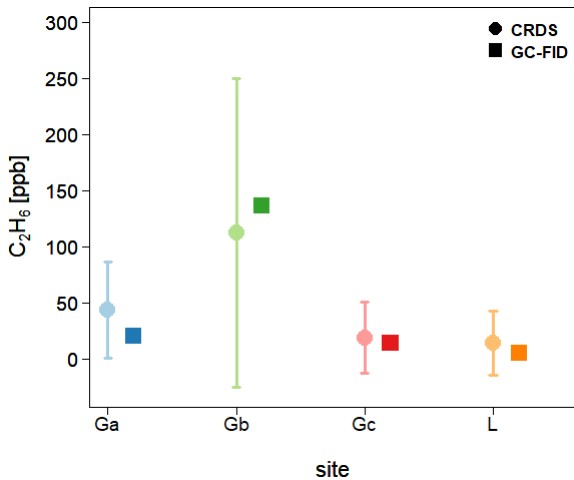

**Figure 8.** Comparison of the $C_2H_6$ mixing ratio measured in situ by CRDS G2201-i and in the laboratory by GC-FID from flask measurements. CRDS G2201-i measurements during the time of flask sampling. Uncertainties (1 SD) are indicated for both CRDS and GC-FID.

between CRDS G2201-i and GC-FID, both installed in the shelter during fieldwork (Assan et al., 2017).

In our study, we went one step further and considered the constraints associated with a mobile setup within a car. As the instrument noise increases during the motion of the car, we decided to stop the car for about 35 min inside the plume to acquire the observations. As it is not possible to stop the car in every place where measurements are made, it is a limitation for this application of the instrument, compared to other instruments able to measure $C_2H_6$ while moving across the plume, such as the LGR UMEA (Lowry et al., 2020) or

an instrument based on the TILDAS method (Smith et al., 2015; Yacovitch et al., 2014, 2020). However, we showed that it is possible to receive reliable values during a short time (e.g., 35 min), and the instrument can be successfully installed inside a vehicle. Notably, having the instrument setup inside the car facilitates the measurement setup as an additional place to install the stationary instrument is not required anymore.

During our tracer release experiment, $C_2H_6 : CH_4$ was calculated from measurements performed when the car was standing inside the plume. With this approach, measured ratios were underestimated. However, using the LGR UMEA instrument, designed for mobile $C_2H_6 : CH_4$ measurements, some discrepancies between the measured and the released value were also observed, albeit smaller. Indeed, in the case of the LGR UMEA measurements, the residuals between measurements and released value were in the range of $-0.015$ to $-0.001$, while the residuals of the CRDS G2201-i are in the range of $-0.018$ to $-0.002$. It is also worth noting that Yacovitch et al. (2014), using a more precise instrument, also reported a systematic underestimation of the $C_2H_6$ mixing ratio by $\sim 6\,\%$.

In our study, during the tracer release experiment, we compared the results obtained by standing stationary inside the plume to sampling air with an AirCore system. The absolute deviation is equal to 0.011 and 0.017 for stationary mode and AirCore mode, respectively. The residuals between released and measured values range from $-0.018$ to $-0.002$ for stationary mode and from $-0.025$ to 0.027 for AirCore mode. Thus, the agreement with released $C_2H_6 : CH_4$ is better for measurements performed by standing inside the plumes than those obtained with the AirCore sampler. During previous studies where CRDS instruments were used (Rella et al., 2015; Lopez et al., 2017), $C_2H_6 : CH_4$ was also

measured using an AirCore sampler. In the study made by Lopez et al. (2017) for pipelines measurements, gas flasks were also collected and measured at INSTAAR (Boulder, CO, USA) using gas chromatography. Based on CRDS measurements with the AirCore sampler, the ethane-to-methane ratio equalled $0.05 \pm 0.01$, while the values achieved from gas chromatography reached $0.04 \pm 0.001$. Overall, the AirCore sampler results were in good agreement with the results from flask measurements.

During the study conducted by Lopez et al. (2017), the CRDS was flushed continuously with a flow rate of $1000\,\mathrm{mL\,min^{-1}}$, controlled by a mass flow controller. During AirCore analysis, the airflow rate was equal to $40\,\mathrm{mL\,min^{-1}}$. This change allowed increasing the number of measurement points by 25 during the replay mode. In our study, in the monitoring mode, we flushed the CRDS instrument with a flow rate of $160\,\mathrm{mL\,min^{-1}}$, and in the replay mode, we increased the number of points only by a factor of 3. These differences could contribute to explaining the discrepancies between measured and released $C_2H_6 : CH_4$ ratios. Further decreasing the flow rate would increase the number of sampling points and could improve the agreement between AirCore-based estimations and actual ratios, especially for small $CH_4$ plumes (e.g., 1–2 ppm above $CH_4$ background). This should be tested to determine the optimal AirCore setup for $C_2H_6 : CH_4$ to improve the characterization of methane sources.

Finally, the $C_2H_6 : CH_4$ ratios obtained by standing inside the plumes are accurate and allow us to separate the different releases at the resolution of the conducted experiment. The results are comparable with results obtained using LGR UMEA, with agreement between measurements and reality also having been confirmed during mobile measurements at gas compressor stations under field conditions. During these measurements, residuals for dry air sampling were between $-0.006$ and $0.009$. Additionally, during fieldwork, flask samples have been taken and measured by GC-FID in the laboratory. During the time of flask sampling at the two gas compressor stations, the $C_2H_6$ mixing ratios were below the value of the instrument CMR ($\sim 50\,\mathrm{ppb}$). For the third gas compressor station, the $C_2H_6$ mixing ratio was above the detection threshold, and the $C_2H_6$ mixing ratio measured by GC-FID was higher than measured by CRDS. Nevertheless, due to the short sampling time of the flasks, these first comparisons between flask samples measured by GC-FID and short-term CRDS field measurements are only approximate, and more comparison campaigns should help to understand the discrepancies between these instruments. In all cases, the standard deviation of $C_2H_6$ measured by the CRDS was close to the averaged value. It shows that the CRDS G2201-i should not be used for measurements of the absolute value of $C_2H_6$ mixing ratios.

Overall, using $C_2H_6 : CH_4$ measured by the CRDS G2201-i, it is possible to separate methane sources between a biogenic origin ($C_2H_6 : CH_4 \sim 0.00$), natural gas leaks and compressors ($C_2H_6 : CH_4 \sim 0.06$; can vary between 0.02–0.17), and processed natural gas liquids ($C_2H_6 : CH_4 \sim 0.3$). $C_2H_6 : CH_4$ of natural gas can vary due to its origin and processing. Also, this instrument can be used to observe the possible temporal variation in $C_2H_6 : CH_4$ of methane emitted from fossil fuel sources. These studies can be made in the vicinity of strongly emitting sources, where $CH_4$ plume reaches at least 1 ppm above background. Determining the exact source of methane inside the industrial site with a lot of potential methane emitters is more challenging to achieve. However, with regards to the results of our study, it is possible to determine the sources of the observed $CH_4$ plume using $C_2H_6 : CH_4$ measured with a CRDS G2201-i if the differences between $C_2H_6 : CH_4$ ratios are higher than 0.01.

## 5 Conclusions and recommendations

The CRDS G2201-i instrument measures $^{12}CO_2$, $^{13}CO_2$, $^{12}CH_4$, $^{13}CH_4$, $H_2O$ and $C_2H_6$, the latter being initially present to correct $^{13}CH_4$ measurements. This study investigates the possibility of performing ethane measurements with a CRDS G2201-i instrument useful for methane source apportionment. The interest is to better constrain methane sources in the laboratory and in the field with two proxies but only one instrument. Before any analysis, $C_2H_6$ raw data must be corrected and calibrated (Fig. 1). The linearity test showed good stability over time, with only a small change in the calibration factors over 4 years. Contrary to the previous studies (Rella et al., 2015; Assan et al., 2017), we do not observe any time drift of the $C_2H_6$ baseline. Nevertheless, regular calibrations and target measurements are recommended.

The controlled release experiment revealed a small systematic underestimation of measured $C_2H_6 : CH_4$ inside the plumes compared to released ones. The larger discrepancy from released $C_2H_6 : CH_4$ occurs in the case of AirCore samplings. Due to that, we recommend standing inside the plumes instead of taking AirCore samples to measure $C_2H_6 : CH_4$ ratios. However, decreasing the flushing flow rate of the CRDS can improve the performance of the instrument during AirCore sampling and should be further investigated in future campaigns.

In this study, we find some limitations of using CRDS G2201-i to measure $C_2H_6 : CH_4$. First of all, we found that we need at least a peak maximum of 100 ppb ethane to gain useful results to help apportion methane sources. Additionally, the required maximum $CH_4$ enhancement above background should be higher than 1 ppm. This threshold is determined using error propagation for a typical $C_2H_6 : CH_4$ ratio equal to 0.1. Under field conditions, this threshold was successfully used for $C_2H_6 : CH_4$ close to 0.05. For weak sources with enhancements below 1 ppm, this limitation prevents $C_2H_6 : CH_4$ measurements from being provided using our approach. Secondly, we have observed significant changes in observed $C_2H_6$ mixing ratios in the presence of

water vapor, and we strongly recommend drying air before making measurements.

Thirdly, due to an increase in the instrument noise during the motion of the car, it is not possible to measure $C_2H_6 : CH_4$ when moving across plumes by car to estimate methane emissions (e.g., Ars et al., 2017). Other "designed for mobile operation" instruments will have to be used in this case for ethane (Yacovitch et al., 2014; Lowry et al., 2020). To work around this problem, $C_2H_6 : CH_4$ can be measured by standing inside the plumes or offline using AirCore sampling after determining the optimal flushing flow (see Sects. 2.2 and 3.2).

Despite these limitations, this study shows the possibility of using the CRDS G2201-i to measure $C_2H_6 : CH_4$ under field conditions with strong methane enhancements, using mobile platforms, and receiving rapid and qualitative results. Even though the instrument is not designed for $C_2H_6 : CH_4$ measurements, after applying correction and calibration factors, when the air is dried, and the methane peak maximum value is at least 1 ppm above background, the CRDS G2201-i gives results that are comparable with released values in controlled experiments and values provided by the gas compressor manufacturing company. Therefore, under these conditions, the CRDS G2201-i instrument can contribute to better constraining methane sources deploying only one instrument, which is possibly already available in the laboratory.

## Appendix A

Rella et al. (2015) quantified the influence of other organic compounds for $\delta^{13}CH_4$ using CRDS G2132, which operates in the same wavelengths as CRDS G2201-i. They also noted that ammonia had a strong influence on ethane. No other compounds from Table 1 (e.g., CO, $CH_3SH$) tested in their paper were noted as having an influence. As CRDS G2132 and CRDS G2201-i operate in the same wavelength, the observed interferences are similar for both instruments.

CRDS G2201-i has the possibility of measuring $H_2S$, $NH_3$ and $C_2H_4$. Similarly to $C_2H_6$ measurements, they are measured to account for their interference for $\delta^{13}CH_4$, and similarly to $C_2H_6$ measurements, they should be calibrated and corrected before any use and before large instrument noise is observed during their measurements. During our study, no signal above instrument noise was observed for $H_2S$, $NH_3$ and $C_2H_4$, so we neglected their interference. Unfortunately, with CRDS G2201-i, it is not possible to measure $C_3H_8$, so we cannot draw conclusions about possible propane interference from our measures. However, as said before, no interference of ethane was noted for propane in Rella et al. (2015). Thus, we assume that propane interference is negligible.

The results, presented in Fig. 3 in the article, were obtained using wet $CH_4$ and $CO_2$ values. In the next step, the analysis of the water vapor sensitivity test was repeated using dry $CH_4$ and $CO_2$ values. These dry values are corrected by default already in the instrument. For all three cases, using dry or wet $CH_4$ and $CO_2$ values did not change the $C_2H_6$ values, which suggests a bigger influence of $H_2O$ than $CH_4$ and $CO_2$ on $C_2H_6$. When the interference correction for low humidity was applied for all steps, the average $C_2H_6$ mixing ratio is equal to $28 \pm 62$ ppb and $28 \pm 61$ ppb for wet and dry $CH_4$ and $CO_2$, respectively. Figure A2 presents a comparison of wet and dry $CO_2$ and $CH_4$ values.

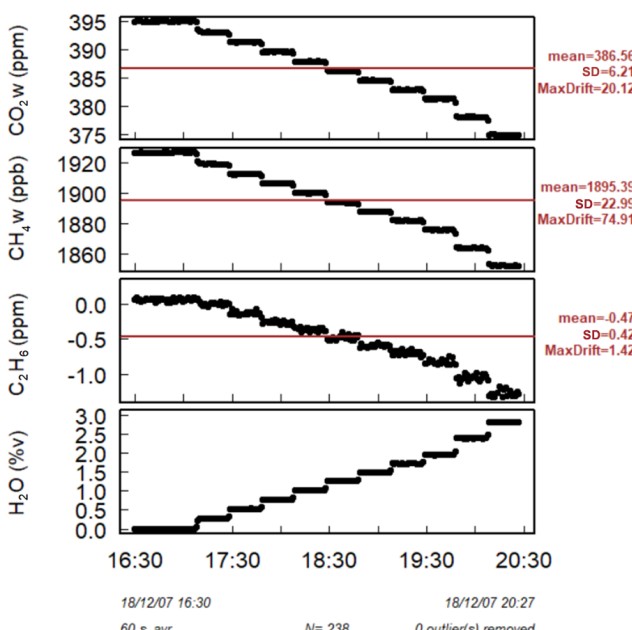

**Figure A1.** $H_2O$ influence on $CO_2$, $CH_4$ and $C_2H_6$.

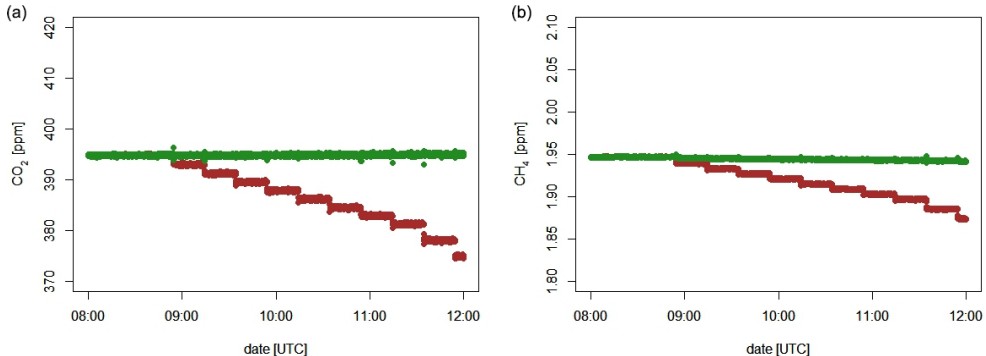

**Figure A2.** Dry (manufactured correction) and wet values of $CO_2$ and $CH_4$. Green: dry values; red: wet values. **(a)** $CO_2$ mixing ratio, **(b)** $CH_4$ mixing ratio.

# Appendix B

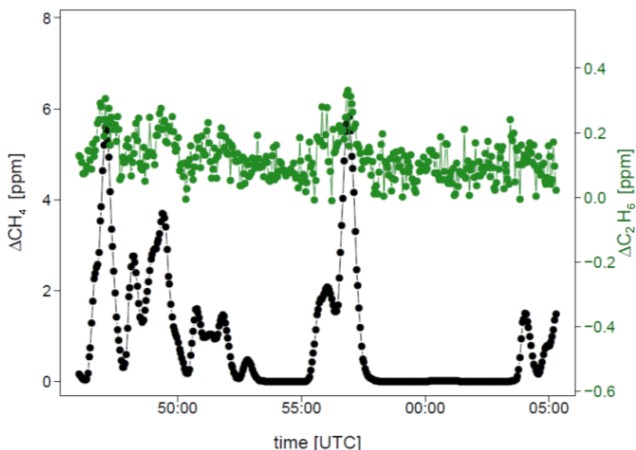

**Figure B1.** $CH_4$ and $C_2H_6$ mixing ratio observed while standing inside the plume.

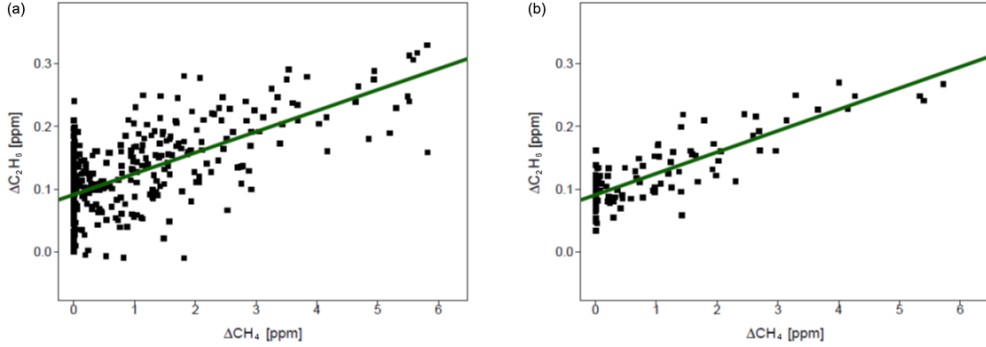

**Figure B2.** $C_2H_6$ mixing ratio vs. $CH_4$ mixing ratio observed while standing inside the plume. **(a)** Non-averaged data, **(b)** 10 s averaged data. Green line: linear fitting.

## Appendix C

During the controlled release experiment (Sects. 2.2 and 3.2), three releases were measured off-site using 5 L bag samples (FlexFoil sample bags, SKC Inc.) filled with air from the plumes. The bag samples were measured afterward in the laboratory without drying. During release 1 and 2, emitted $C_2H_6 : CH_4$ was equal to 0.00, the third release having a $C_2H_6 : CH_4$ of about 0.032. In all cases, for background samples, the $C_2H_6$ mixing ratio was found higher than for the bag samples collected inside the plumes. Due to that, results from the bag samples are rejected from further analysis. There are two possible reasons for the incorrect values obtained with bag samples. Firstly, these bags could not be adapted for storing ethane. Secondly, as the samples were wet, the $H_2O$, $CO_2$ and other species interferences on $C_2H_6$ could be higher and not linear. Thus, the applied interference correction did not improve the measured $C_2H_6$ mixing ratio.

**Table C1.** $C_2H_6 : CH_4$ with interference correction for high humidity.

| ID | $CO_2$ [ppm] | $CH_4$ [ppm] | $\delta^{13}CH_4$ [‰] | $H_2O$ [%] | $C_2H_6$ [ppm] | $C_2H_6 : CH_4$ [ppm ppm$^{-1}$] |
|---|---|---|---|---|---|---|
| 1.1b | 402 | 2.23 | −47 | 1.25 | $0.27 \pm 0.06$ | $0.12 \pm 0.03$ |
| 1.2b | 397 | 2.01 | −47 | 1.22 | $0.27 \pm 0.06$ | $0.13 \pm 0.03$ |
| 1.3b | 399 | 3.34 | −45 | 1.22 | $0.39 \pm 0.06$ | $0.12 \pm 0.02$ |
| 1.4b* | 395 | 1.96 | −48 | 1.23 | $0.44 \pm 0.06$ | $0.22 \pm 0.03$ |
| 1.5b | 399 | 2.31 | −46 | 1.29 | $0.43 \pm 0.06$ | $0.19 \pm 0.03$ |
| 1.6b | 399 | 5.25 | −43 | 1.29 | $0.45 \pm 0.07$ | $0.09 \pm 0.01$ |
| 1.7b | 402 | 5.19 | −44 | 1.29 | $0.62 \pm 0.09$ | $0.12 \pm 0.02$ |
| 1.8b* | 396 | 1.98 | −48 | 1.25 | $0.55 \pm 0.08$ | $0.28 \pm 0.04$ |
| 2.1b | 420 | 3.25 | −45 | 1.27 | $0.55 \pm 0.07$ | $0.17 \pm 0.02$ |
| 2.2b* | 397 | 1.97 | −49 | 1.17 | $0.72 \pm 0.15$ | $0.36 \pm 0.08$ |

* Background samples.

**Appendix D**

Comparison of raw data and 10 s averaged data from measurements in the Île-de-France region.

**Table D1.** Fieldwork analysis. Ga, Gb and Gc: gas compressor; L: landfill;

| ID | Max $\Delta CH_4$ | Max $\Delta C_2H_6$ | 1 s | $r^2$ | 10 s | $r_2$ | $n$ | Date (dd.mm.yyyy) |
|---|---|---|---|---|---|---|---|---|
| Ga1[a] | 1.486 | 0.309 | $0.070 \pm 0.013$ | 0.162 | $0.066 \pm 0.018$ | 0.235 | 138 | 16.05.2019 |
| Ga2 | 1.737 | 0.269 | $0.060 \pm 0.005$ | 0.195 | $0.059 \pm 0.007$ | 0.303 | 533 | 16.05.2019 |
| Ga3 | 5.85 | 0.414 | $0.045 \pm 0.002$ | 0.489 | $0.044 \pm 0.003$ | 0.645 | 495 | 15.07.2019 |
| Gb1[a] | 7.314 | 0.878 | $0.090 \pm 0.001$ | 0.852 | $0.091 \pm 0.002$ | 0.927 | 811 | 27.05.2019 |
| Gb2[a] | 0.513 | 0.323 | $0.085 \pm 0.022$ | 0.024 | $0.083 \pm 0.029$ | 0.044 | 594 | 12.07.2019 |
| Gb3 | 1.454 | 0.26 | $0.052 \pm 0.007$ | 0.082 | $0.05 \pm 0.009$ | 0.15 | 613 | 12.07.2019 |
| Gb4 | 1.677 | 0.236 | $0.046 \pm 0.008$ | 0.086 | $0.05 \pm 0.011$ | 0.174 | 336 | 12.07.2019 |
| Gc1[b] | 0.495 | 0.284 | $0.091 \pm 0.037$ | 0.037 | $0.09 \pm 0.021$ | 0.082 | 711 | 28.05.2019 |
| L | 1.516 | 0.266 | $0 \pm 0.006$ | 0 | $0 \pm 0.007$ | 0 | 712 | 16.05.2019 |

[a] A1, B1 and B2 rejected from further analysis (wet air) and [b] C1 rejected from further analysis (low enhancement). Raw and 10 s averaged data. CE3 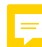

*Data availability.* Data from the fieldwork and most of the laboratory tests are available on the Carbon Portal (https://doi.org/10.18160/hr89-ks44, Defratyka, 2021). Data from the time drift test are available on request.

*Author contributions.* SMD wrote the article and codes to plot figures. All co-authors reviewed and edited the primary version of the article. SMD conducted and analyzed laboratory tests described in Sect. 2.1.1 and 2.1.4, prepared and conducted fieldwork measurements in the IDF region, and conducted measurements by CRDS during controlled release experiments. DL conducted and analyzed the laboratory test described in Sect. 2.1.2 and 2.1.3, actively participated in fieldwork, collected flask samples, and analyzed data from fieldwork in the IDF region. VG provided analysis of flask samples by GC-FID. JF, JH and NY organized the controlled release experiment. JH and NY provided analysis of mixture gas released during the controlled release experiment. JF analyzed data from LGR UMEA during the controlled release experiment. JDP, CYK and PB helped SMD with formal analysis, helped conduct laboratory test and mobile measurements, and helped develop scientific questions behind the article.

*Competing interests.* The authors declare that they have no conflict of interest.

*Acknowledgements.* We acknowledge our laboratory colleagues Carole Philippon and Luc Lienhardt for sharing results of tests conducted at the ATC MLab. We also gratefully thank Dominique Baisnee for the measurements of flask samples on the GC-FID. We gratefully acknowledge the GRTgaz company for sharing data with us and helping to improve the manuscript, especially Pascale Guillo-Lohan, Pascal Alas, Francis Bainier and Jean-Luc Fabre.

*Financial support.* This research has been supported by the European Union's Horizon 2020 research and innovation program (Marie Skłodowska-Curie grant no. 722479).

*Review statement.* This paper was edited by Glenn Wolfe and reviewed by three anonymous referees.

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

**Remarks from the language copy-editor**

CE1    Please give an explanation of why this needs to be changed. We have to ask the handling editor for approval. Thanks.

CE2    Please give an explanation of why this needs to be changed. We have to ask the handling editor for approval. Thanks.

CE3    Please give an explanation of why this needs to be changed. We have to ask the handling editor for approval. Thanks.