# Peer review of "Ethane measurement by Picarro CRDS G2201-i in laboratory and field conditions: potential and limitations"

_Atmospheric Measurement Techniques, 2020_

## Referee Comment (RC1) · Anonymous Referee #1 · 14 Nov 2020

The stated main objective of this paper: is to evaluate the performance of the CRDS G2201-i and the applicability of making short-term, direct, continuous, mobile measurements of ethane in methane-enriched air, with sufficient precision during near source ("pollution plume conditions") surveys. The authors did a commendable amount of work characterizing their instrument and in presenting all the limitations of this instrument. This is to their credit, and to this end the work described herein achieves its objectives.

However, with that being said, this reviewer finds very limited applications where the G2201-i analyzer can be employed in measuring ethane/methane slopes in real world

situations. As stated, peak ethane enhanced values of at least 100 ppb and peak methane values of at least 1 ppm are needed on stationary platforms for meaningful slopes. Unless one is directly at a well head or at a compressor source, this type of performance is not very useful. Also, more discussions of the 30 ppb bias in their ethane measurements, its sources, and its variability are needed.

Despite the body of work here, this reviewer does not find any utility in publishing this paper with very limited real world applications for the G2201-i analyzer in terms of ethane/methane slopes. This reviewer recommends that the authors instead focus on a similar concerted effort to characterize their CRDS 2210-i analyzer, which they briefly mention, and shows superior performance for ethane.

---

## Referee Comment (RC2) · Anonymous Referee #2 · 25 Nov 2020

Defratyka et al. report ethane measurements from a spectroscopic instrument originally not designed to make an ethane measurement. Ethane has a small interfering absorption peak for an instrument that reports isotopic measurements of CO2 and CH4. Defratyka et al. quantify this peak, and although the ethane measurement has low precision, use it to quantify the ethane to CH4 ratio from natural gas emissions.

Although the ethane measurement is not very good, and the application of this measurement is limited, it nonetheless could be of some use to the scientific community. However, before this paper is ready for publication, I think some issues must be addressed.

[Figure]

The Picarro 2201 website (https://www.picarro.com/sites/default/files/product_documents/Picarro_%20G2201-i%20Analyzer%20Datasheet_053017.pdf) says there are interferences from "other organics", as well as ethane, ammonia, ethylene, and sulfur-containing compounds. Might some of these other organic exist in natural gas? Have the authors looked at propane interferences?

I was confused in the second sentence of the Abstract by the use of the word "dedicated", which is also used throughout the paper. To me, "dedicated" means it only measures that to which it is dedicated, in this case $CO_2$ and $CH_4$. I suggest the authors use "originally designed to measure" or some such phrase. And it wasn't until line 84 on page 3 that the authors mention for the first time that the G2201-i was actually used to measure ethane. The authors should explicitly state that the G2201-i was used to measure ethane in the Abstract, rather than hiding it in terms of "consider[ing] the possibility" of measuring ethane.

Generally, I thought the paper needed more statements of introduction and conclusion in many paragraphs. There are a lot of paragraphs explaining what the authors did related to the measurement. What is missing is information on why they are doing this, and what are the results of this part of the experiment.

I also would like to see this paper act more as a stand-alone work. As written, it is tied heavily to Assan et al. (2017) in too many places. In many cases, a sentence or two summarizing the results of the cited work would be helpful.

Other comments:

line 36, somewhere it should be stated that the ratios referred to in the paper are molar ratios, as opposed to mass ratios

line 58, instead of simply stating "good agreement", add what measurements agreed well in case the reader is not familiar with Assan et al. (2017)

line 66, again, please list the measurements that were compared

line 88, Equation 1: rather than a generic equation, please fill in the parameters A, B, and C so that one does not have to look at the paper by Assan et al. to find these numbers

line 97. Agreed. What have the authors done to ensure comparability and traceability?

line 105, if CMR is commonly known as precision, why not use the phrase precision?

line 153, vibrations of the instrument probably lead to instrument noise regardless of whether then instrument is "dedicated" to an ethane measurement. And are the authors referring to the ethane measurement noise when referring to "instrument readouts"? Or all measurements? And in line 154, this is referred to as a "constraint". Does this mean the mobile data were noisy to the point of being unusable?

line 162, are these two-sided fits? Weighted by anything?

line 174, what is "skc"?

line 181, where has this publication been submitted? Is it available to read?

line 185, what is the purpose of this sentence? Was the change in drying intentional? If so, for what reason? Was it regular?

line 201, the authors should define a "low" amount of ethane. It seems like they are referring to 23 ppb, which is not low. But reading later, it appears they are referring to 2.2 ppb? But that is in the next section, so I'm not sure if that is the same working standard referred to in this section. Regardless, the authors should start with their best estimate of the tank mixing ratio. This puts the G2201-i performance in perspective. Otherwise, the reader has to read several paragraphs to discover a 2.2 ppb standard reads as 23 ppb on this instrument.

For Table 1 and Figure 1, Was this the working gas used as part of the dilution system described on page 4/equation 1? In general, I think whenever the authors mention a working gas, they should state what the nominal ethane mixing ratio is.

line 210, can you add an uncertainty to the 2.2 ppb?

line 212, this was a surprisingly high offset. I am also still getting confused by the working gases used. This is apparently not the same one used for Figure 1? And if Figure 1 averaged 23 ppb, presumably you were giving it less than 2.2 ppb (2.2 * 23/33 = ~1.5 ppb?).

line 218 and 220, there are a lot of Picarro model numbers in this paragraph. Perhaps the authors could add a table to show what models measure what species, since I am not familiar with all the models.

Figure 2, what units are the Allan deviation plots in? I assume they are all ppm?

line 240, in some cases such as these, a standard error of the mean would also be worth reporting, along with the standard deviation

Figure 3, are the differences between Protocol 2 and 3 simply linear fits, i.e., Protocol 2 fits a line to all the data, and Protocol 3 fits a line to data < 0.16% H2O? If so, what would a higher-order fit to H2O do – could you use that for both high and low humidity cases? I'm also not sure of the benefit of naming these "Protocol X", since every time they are mentioned, a description of the Protocol is also given. It seems easier to mention "no correction", etc. every time, and the reader wouldn't have to remember what arbitrary Protocol number this was given.

line 255, what does a release with a C2H6:CH4 equal to 0 mean? No ethane was released, but methane was? Or nothing at all was released?

line 264, when absolute deviations are on the order of 10 ppb, an "improvement" of 0.4 ppb seems like simple statistical variation. In other words, I think the authors are assigning significance to the insignificant digits of these numbers.

Table 3, Why do the authors report the residuals, and not the ratio itself? And how are the residuals defined? Is a linear fit performed on the data, and these are the residuals when the fitted line is subtracted from the data?

[Figure]

Table 4, referring to the different sites as A, B, C, and D only further complicates this table. Also, I'm not yet sure what difference the survey number makes. I think it would be easier to refer to these as compressor 1, 2, 3, and landfill. Use abbreviations if necessary. Also, move the * information from the title of the table to below the table.

Figure 5, how are these slopes calculated? Are the data weighted in the fit? And are the uncertainties reported in Table 4 just the slope uncertainties, or do they tie in the uncertainties of the C2H6 measurement?

line 407, it might be best to reiterate the requirement that CH4 be greater than 1 ppm here, as mentioned previously in the paper

Grammar suggestions/typos:

line 28, it looks like "sources" is possessive, needs apostrophe

line 34 and elsewhere, I think "ethane:methane ratio" is redundant. Suggest either "ethane:methane" or "the ethane to methane ratio". But to me, using a colon implies ratio.

line 42–43, change "methane enhancement source" to "methane source"

line 45, remove "access to"

line 56, change "biogenic or thermogenic" to "biogenic from thermogenic"

line 85, add "1" to H in CH4 for consistency

line 124, change to "Equation"

line 133, change "has been measured during" to "was sampled for"

line 152, change "the previous works" to "previous work"

line 155, change "standing some" to "spending"

line 155, change "accumulating air in" to "sampling air using"

line 159, add "the" before "C2H6:CH4"

lines 164–5, start sentence "A description of the experimental. . .", replace "find" with "found", and add period after "(2017)"

line 167, suggest "up to" instead of "until"

line 169, suggest "C2H6:CH4" instead of "ethane:methane" for consistency

line 170, suggest "stationed in the plume"

line 171–172, suggest ". . . the time spent within the plume was approximately 15 to 20 minutes."

line 173, suggest "tracer release"

line 174, change "5 liters'" to "5-liter". Also line 284.

line 175, change to ". . . bags were sampled inside . . . and one was sampled . . ."

line 175, change "bags" to "bag samples"

line 180, delete "real". I think "field" is sufficient.

line 183, add "the" before "C2H6:CH4"

lines 185–186, change "part of measurements without dryer" to either "part with a dryer" or "part of the measurements without a dryer"

Figure 1 caption, I would re-word and make two sentences, change "20 minutes" to "20-minute", start new sentence with "For each measurement point, squares represent. . ."

line 214, change to "As a result. . ."

line 218, change "dedicated to the measure of ethane" to "designed to measure ethane"

line 227, change "ethane absolute value" to "an absolute value of ethane"

line 229, change "deduct" to "deduce"

line 259 and elsewhere, suggest "stationary in-plume situation" instead of "plume standing situation"

line 273, suggest something like: "For the higher emission, the measurements and results were combined when the emission rates were 70, 72, and 73 L/min."

line 276, add "the" before "AirCore"

line 285–286, add "to" after "equal"

Table 4, change "Data" to "Date"

line 321, change "due the very" to "due to the very"

line 329, change "ratio" to "ratios"

line 356–357, the time of sampling is confusing. The first sentence makes it sound like the instrument spends 10 minutes online, followed by 10 minutes offline. The next sentence makes it sound like the instrument spends 10 minutes online, followed by 20 minutes offline.

line 358, perhaps just describe the CRDS data as being averaged over the sampling time of the GC-FID

line 360–361, change to "... to use a CRDS G2201-i to measure C2H6:CH4, ..."

line 366, change to "... on the TILDAS method ..."

line 367, change to "tracer release"

line 384–385, a word is missing here, perhaps "allowed us to", change "measurements point" to "measurement points"

line 391, change to either "allow us to separate" or "allow the separation"

line 394, change to "flask samples"

line 398, indicative of what?

[Figure]

for the tables in the Appendix, I would put the * asides below the table, rather than part of the table title

[Figure]

---

## Referee Comment (RC3) · Anonymous Referee #3 · 2 Dec 2020

General Comments

This manuscript assesses the ethane measurement obtained using the Picarro G2201-i and tests its ability to provide meaningful data for determining C2H6:CH4 in methane plumes, with the goal of source attribution. The instrument is tested and calibrated in the laboratory, subjected to controlled release experiments, and taken to measure real sources in the field. The authors find that, due mostly to the low precision of the ethane measurement ($\sim$ 50 ppb), the G2201-i can only realistically be used for ethane-to-methane ratios in methane peaks that are at least 1 ppm above the background. Furthermore, the measurement as presented must be taken under stationary

conditions (i.e., with the mobile platform parked within a plume for ~30 min) or the noise of the ethane measurement becomes unacceptably high.

The use of the G2201-i for the described applications seems extremely limited, especially in light of the other available instruments that can do this type of measurement much better (LGR, Aerodyne, and other Picarro models). However, the authors do recognize that in order to use the Picarro G2201-i for ethane field measurements (which in turn are to be used only in the calculation of ethane-to-methane ratios rather than absolute ethane mixing ratios), the instrument response must be extensively characterized. This work is done, and the limitations of the G2201-i for the purposes described are appropriately determined and discussed.

There is a lot of information presented on the experimental details of previous work, which, in my opinion, obscures the experimental design and the main conclusions of the current manuscript somewhat. It makes it difficult for the reader to focus on the important points of the manuscript (one of which is the many conditions that need to be satisfied to obtain useful ethane information from the G2201-i). I recommend the authors try to streamline the manuscript as much as possible so that the important points are evident. Additionally, I recommend careful proofreading of the manuscript, which contains many small grammar errors, some of which are highlighted below under "Technical Comments".

Specific Comments

Lines 53-58: How does this study differ from Assan et al? Is the system just characterized better? Is the only difference, as mentioned later in lines 361+, that the instrument was put in a car (which must remain stationary within a plume for ~30 mins to take a useful measurement)? If so, that should be made clear early on.

Lines 62+: Did you use the monitoring mode in addition to the replay mode for the Aircore in the current study? I think some more information on how the Aircore was used specifically for this study should be included, although I would add this information

later in the methods section.

Lines 82 – 85: This background information on how ethane is measured and reported for an isotopic methane/carbon dioxide instrument should be moved to the abstract and introduction.

Lines 81+ (Materials and Methods section): To make each factor investigated clear, consider re-formatting with subheadings, such as, 1.1 Laboratory 1.1.1 Interference Correction on Ethane and Water Sensitivity 1.1.2 Ethane Calibration Factors 1.1.3 Precision and Allan Variance 1.1.4 Time drift Because the water vapor sensitivity tests are tests on the validity of the interference corrections, I think this should be discussed at the same time as the interference correction in general.

Lines 147+: I have some confusion about what Protocol 1, 2, and 3 are. Are these described clearly somewhere? I would add relevant details here in the methods section.

Lines 151-152: Delete "The measurement setup used here is the same as in the field" and only mention in section 2.3.

Lines 154-159: The point that true "mobile" measurements are not conducted (i.e., while the vehicle is moving) should be highlighted earlier in the manuscript. It is an important point that is somewhat hidden here. Also- please add information here about the specifics of the Aircore setup as used in this study (e.g., flow rates, different modes, car stopped or moving).

Line 174- 178: I question whether any of the information about the failed bag measurements should be included in the main manuscript, especially given the issues with sampling and bag preparation mentioned later (in lines 284+). Maybe make a very abbreviated reference to them, and then move all other bag information to the supporting information.

Lines 195+, Section 3.1: Suggest headings that are the same as those suggested above for section 2.1 to help organize the information.

Line 201: Can you specify what a "low amount of C2H6" means?

Line 356: Please clarify what "10 mins of ambient air collection was measured during 20 minutes" means.

Lines 384-390: Please revise this section on Aircore and CRDS flow rates for clarity. How are the Aircore and CRDS flows related? Were there reasons for the chosen flows?

Line 398: Do you mean the "first comparisons" of ethane mixing ratios with GC-FID match up in a relative sense? The word "indicative" is confusing here.

Line 431: Please clarify what the "flushing issue" to be solved is.

Technical Comments

Line 43: "source" should be "sources"

Line 54: "measure of" should be "measurement of"

Line 60: Change "allows to improve time resolution" to "allows improvement of time resolution"

Line 77: Change "instrument to ethane" to "instrument for ethane"

Line 165: Change "find" to "found"

Line 167: Change "emission flux" to "gas flow rate"

Line 168: Change "could vary" to "were varied"

---

## Author Comment (AC1) · 5 Feb 2021

Author Comments on the manuscript 10.5194/amt-2020-410-RC1, Reviewer 1

We would like to thank Reviewer 1 for his comments that helped us to improve our manuscript. We provide below a detailed reply to the Reviewer's comment on the utility of our work. Manuscript will be clarified accordingly.

The stated main objective of this paper: is to evaluate the performance of the CRDS2201-i and the applicability of making short-term, direct, continuous, mobile measurements of ethane in methane-enriched air, with sufficient precision during near

source ("pollution plume conditions") surveys. The authors did a commendable amount of work characterizing their instrument and in presenting all the limitations of this instrument. This is to their credit, and to this end the work described herein achieves its objectives. However, with that being said, this reviewer finds very limited applications where the G2201-i analyzer can be employed in measuring ethane/methane slopes in real world situations. As stated, peak ethane enhanced values of at least 100 ppb and peak methane values of at least 1 ppm are needed on stationary platforms for meaningful slopes. Unless one is directly at a well head or at a compressor source, this type of performance is not very useful. Also, more discussions of the 30 ppb bias in their ethane measurements, its sources, and its variability are needed. Despite the body of work here, this reviewer does not find any utility in publishing this paper with very limited real world applications for the G2201-i analyzer in terms of ethane/methane slopes. This reviewer recommends that the authors instead focus on a similar concerted effort to characterize their CRDS 2210-i analyzer, which they briefly mention, and shows superior performance for ethane.

A:We think that the full characterization of CRDS 2201-i analyzer to measure ethane to methane ratios proposed in our paper is useful and worth publishing for the following reasons: 1. Valuable opportunities exist for using this instrument beyond its intended application (i.e. measuring ethane together with isotopic composition on a single analyzer), but this requires a prior specific characterization. In our study we focused on the characterization of CRDS G2201-i for ethane measurements, as some previous studies already used this instrument during field campaigns to measure ethane to methane ratio in fixed settings such as a shelter (described in paragraph 4 discussion). Thus our purpose was to evaluate limitations and possibilities to use this instrument to measure ethane to methane ratio in a car setting (one conclusion was that, indeed, it needs to be stationary during measurements but is mobile over a site). This study is useful for other scientific teams, which do not have an instrument dedicated for ethane measurements, but already have the CRDS G2201-i and would like to use it in field conditions for measuring both $\delta$13CH4 and ethane to methane ratio. According to our

knowledge, outside our team, two other research teams in Europe use CRDS G2201-i during mobile measurements (Heidelberg University and AGH University of Science and Technology). Possibly, more institutions use it as well. Thus, our manuscript can be viewed as a protocol where all necessary steps are described and verified before field work. On a side note, the CRDS 2210-i briefly discussed was only tested in our colleagues' laboratory but was on loan from another institute. It was not available for us to test in field conditions.

2. Large enhancements necessary to use CRDS G2201-i can be found for many sources. As you note, using CRDS G2201-i to calculate ethane to methane ratio requires relatively large CH4 and C2H6 enhancement. Indeed, these conditions do not happen often during long-term stationary measurements on a fixed station located remotely from sources. However, high enhancements are observed near-source surveys for most types of methane point or site-scale sources like coal mines, natural gas or oil, waste water treatment plants, landfills, geological sources (e.g. Zazzeri et al. 2015; Lopez et al. 2017; Hoheisel et al. 2019; Lowry et al. 2020). Moreover, recent studies (Lan et al. 2019; Turner, Frankenberg, and Kort 2019; Yacovitch, Daube, and Herndon 2020) showed varying ethane to methane ratios for different facilities, even at a local scale, which shows the important role of near-source measurements of ethane to methane ratio. Having an additional model of analyzer to measure this ratio increase the possibilities to perform systematic repetitions of these measurements that can be used to observe possible changes of ratios over time

3. Ethane to methane ratios are important to better estimate methane sources from different emitting processes and every instrument counts in this matter. Mobile near-source measurements of C2H6:CH4 ratio also allows for partitioning sources between biogenic (e.g., landfill, farms) and thermogenic (e.g., oil and natural gas facilities) on a small scale, as biogenic sources do not co-emit ethane (Yacovitch et al. 2014; Assan et al. 2017). So far, to achieve it, $\delta 13CH4$ is commonly used, as typically, biogenic sources are more depleted than thermogenic sources (Nisbet et al. 2019; Turner,

Frankenberg, and Kort 2019; Saunois et al. 2020). However, recent studies showed that some fossil fuel sources can also emit more depleted CH4 (Schwietzke et al. 2016; Sherwood et al. 2017; Yacovitch, Daube, and Herndon 2020). These more depleted 13C values are caused by the biogenic origin of the extracted gas. Based on the current database, 14% of conventional natural gas samples have a biogenic origin ($\delta$13CH4 < -55‰ (Sherwood et al. 2017). In this case, it is crucial to use an additional tracer to portion CH4 sources during mobile near-source measurements. For this purpose, C2H6:CH4 measurements can be performed during mobile near-source surveys and using existing instruments in the different networks provides a bonus instead of waiting for their replacement with more recent and accurate instruments (e.g. 2210-i). Considering the crucial role of near-source measurements of ethane to methane ratio, our method, even with some limitations, can give rapid and qualitative results to determine the origin of methane emission.

In the revised version of the manuscript we will discuss in details the observed 30 ppb bias, which comes from the instrument various terms of uncertainty. We will also improve the introduction and discussion part to highlight the opportunities arising from our research for the scientific community and we will make more clear the motivation of our study.

References

Assan, Sabina, Alexia Baudic, Ali Guemri, Philippe Ciais, Valerie Gros, and Felix R. Vogel. 2017. "Characterization of Interferences to in Situ Observations of Delta13CH4 and C2H6 When Using a Cavity Ring-down Spectrometer at Industrial Sites." Atmospheric Measurement Techniques 10 (6): 2077–91. https://doi.org/10.5194/amt-10-2077-2017. Hoheisel, Antje, Christiane Yeman, Florian Dinger, Henrik Eckhardt, and Martina Schmidt. 2019. "An Improved Method for Mobile Characterisation of $\Delta$13CH4 Source Signatures and Its Application in Germany." Atmospheric Measurement Techniques 12 (2): 1123–39. https://doi.org/10.5194/amt-12-1123-2019. Lan, Xin, Pieter Tans, Colm Sweeney, Arlyn Andrews, Edward Dlugokencky, Stefan Schwietzke,

Jonathan Kofler, et al. 2019. "Long‐Term Measurements Show Little Evidence for Large Increases in Total U.S. Methane Emissions Over the Past Decade." Geophysical Research Letters 46 (9): 4991–99. https://doi.org/10.1029/2018GL081731. Lopez, M., O.A. Sherwood, E.J. Dlugokencky, R. Kessler, L. Giroux, and D.E.J. Worthy. 2017. "Isotopic Signatures of Anthropogenic CH4 Sources in Alberta, Canada." Atmospheric Environment 164 (September): 280–88. https://doi.org/10.1016/j.atmosenv.2017.06.021. Lowry, David, Rebecca E. Fisher, James L. France, Max Coleman, Mathias Lanoisellé, Giulia Zazzeri, Euan G. Nisbet, et al. 2020. "Environmental Baseline Monitoring for Shale Gas Development in the UK: Identification and Geochemical Characterisation of Local Source Emissions of Methane to Atmosphere." Science of The Total Environment 708 (March): 134600. https://doi.org/10.1016/j.scitotenv.2019.134600. Nisbet, E. G., M. R. Manning, E. J. Dlugokencky, R. E. Fisher, D. Lowry, S. E. Michel, C. Lund Myhre, et al. 2019. "Very Strong Atmospheric Methane Growth in the 4 Years 2014–2017: Implications for the Paris Agreement." Global Biogeochemical Cycles 33 (3): 318–42. https://doi.org/10.1029/2018GB006009. Saunois, Marielle, Ann R. Stavert, Ben Poulter, Philippe Bousquet, Joseph G. Canadell, Robert B. Jackson, Peter A. Raymond, et al. 2020. "The Global Methane Budget 2000–2017." Preprint. Atmosphere – Atmospheric Chemistry and Physics. https://doi.org/10.5194/essd-2019-128. Schwietzke, Stefan, Owen A. Sherwood, Lori M. P. Bruhwiler, John B. Miller, Giuseppe Etiope, Edward J. Dlugokencky, Sylvia Englund Michel, et al. 2016. "Upward Revision of Global Fossil Fuel Methane Emissions Based on Isotope Database." Nature 538 (7623): 88–91. https://doi.org/10.1038/nature19797. Sherwood, Owen A., Stefan Schwietzke, Victoria A. Arling, and Giuseppe Etiope. 2017. "Global Inventory of Gas Geochemistry Data from Fossil Fuel, Microbial and Burning Sources, Version 2017." Earth System Science Data 9 (2): 639–56. https://doi.org/10.5194/essd-9-639-2017. Turner, Alexander J., C. Frankenberg, and Eric A. Kort. 2019. "Interpreting Contemporary Trends in Atmospheric Methane." Proceedings of the National Academy of Sciences 116 (8): 2805–13. https://doi.org/10.1073/pnas.1814297116. Yacovitch, Tara I., Conner Daube, and Scott C. Herndon. 2020. "Methane Emissions from Offshore

Oil and Gas Platforms in the Gulf of Mexico." Environmental Science & Technology 54 (6): 3530–38. https://doi.org/10.1021/acs.est.9b07148. Yacovitch, Tara I., Scott C. Herndon, Joseph R. Roscioli, Cody Floerchinger, Ryan M. McGovern, Michael Agnese, Gabrielle Pétron, et al. 2014. "Demonstration of an Ethane Spectrometer for Methane Source Identification." Environmental Science & Technology 48 (14): 8028–34. https://doi.org/10.1021/es501475q. Zazzeri, G., D. Lowry, R.E. Fisher, J.L. France, M. Lanoisellé, and E.G. Nisbet. 2015. "Plume Mapping and Isotopic Characterisation of Anthropogenic Methane Sources." Atmospheric Environment 110 (June): 151–62. https://doi.org/10.1016/j.atmosenv.2015.03.029.

---

## Author Comment (AC2) · 5 Feb 2021

Author comments on the manuscript 10.5194/amt-2020-410-RC2, Reviewer 2

We would like to thank Reviewer 2 for the constructive comments that aided us to improve our manuscript. In this document, we provide our replies to the Reviewer's comments. Following every comment, we give our reply. Here line numbers, page numbers and figure numbers refer to the original version of the manuscript.

Defratyka et al. report ethane measurements from a spectroscopic instrument originally not designed to make an ethane measurement. Ethane has a small interfering

absorption peak for an instrument that reports isotopic measurements of CO2 and CH4. Defratyka et al. quantify this peak, and although the ethane measurement has low precision, use it to quantify the ethane to CH4 ratio from natural gas emissions. Although the ethane measurement is not very good, and the application of this measurement is limited, it nonetheless could be of some use to the scientific community. However, before this paper is ready for publication, I think some issues must be addressed.

1. The Picarro 2201 website (https://www.picarro.com/sites/default/files/product_documents/Picarro_%20G2201-i%20Analyzer%20Datasheet_053017.pdf) says there are interferences from "other organics", as well as ethane, ammonia, ethylene, and sulfur-containing compounds. Might some of these other organic exist in natural gas? Have the authors looked at propane interferences?

A:Rella et all (2015) quantified the influence of other organic compounds for $\delta$13CH4 using CRDS G2132, which operates in the same wavelengths as CRDS G2201-i. They also noted that ammonia was having a strong influence on ethane. No other organic compounds from Table 1 tested in their paper were noted as having an influence. As CRDS G2132 and CRDS G2201-i operate in the same wavelength, the observed interferences are similar for both instruments. CRDS G2201-i has the possibility to measure H2S, NH3 and C2H4. Similarly, to C2H6 measurements, they are measured to account for their interference for $\delta$13CH4 and, similarly to C2H6 measurements, they should be calibrated and corrected before any use and large instrument noise is observed during their measurements. During our study, no signal above instrument noise was observed for H2S, NH3 and C2H4 so we neglected their interference. Unfortunately, with CRDS G2201-i, it is not possible to measure C3H8, so we cannot conclude about possible propane interference from our measures. However, as said before, no interference on ethane was noted for propane in Rella et al. Thus, we assume that propane interference is negligible.

2. I was confused in the second sentence of the Abstract by the use of the word

"dedicated", which is also used throughout the paper. To me, "dedicated" means it only measures that to which it is dedicated, in this case, CO2 and CH4. I suggest the authors use "originally designed to measure" or some such phrase. And it wasn't until line 84 on page 3 that the authors mention for the first time that the G2201-i was actually used to measure ethane. The authors should explicitly state that the G2201-i was used to measure ethane in the Abstract, rather than hiding it in terms of "consider[ing] the possibility" of measuring ethane.

A:In the revised version, "Dedicated" will be changed into "originally designed to measure" in abstract and in the rest of the manuscript. Also, it will be clearly said in the abstract and introduction that CRDS G2201-i was used to measure C2H6.

3. Generally, I thought the paper needed more statements of introduction and conclusion in many paragraphs. There are a lot of paragraphs explaining what the authors did related to the measurement. What is missing is information on why they are doing this, and what are the results of this part of the experiment.

A:We will improve the introduction to highlight importance of mobile, near source mobile measurements. Also results and conclusions of every part of experiment will be detailed and explained. Overall application and significance of work will be described.

4. I also would like to see this paper act more as a stand-alone work. As written, it is tied heavily to Assan et al. (2017) in too many places. In many cases, a sentence or two summarizing the results of the cited work would be helpful.

A:The method section will be rewritten to make it clearer and complete, to be self-independent. Below equation 1, table with values of factors A, B, C for different water vapor levels will be added. Also, a scheme of necessary steps to calibrated and correct C2H6 will be added. The set-up of linearity test will be also added.

Other comments: 5. line 36, somewhere it should be stated that the ratios referred to in the paper are molar ratios, as opposed to mass ratios

A:The sentence will be added: Based on mobile measurements, as CH4 and C2H6 mixing ratios are measured, ethane to methane ratio is calculated as molar ratio.

6. line 58, instead of simply stating "good agreement", add what measurements agreed well in case the reader is not familiar with Assan et al. (2017)

A:The sentence will be added: Ethane:methane ratio from flask samples allowed to distinguish methane emissions from the two pipelines. The natural gas in pipeline 1 had ratio equaled to 0.074 ± 0.001 ppm C2H6/ppm CH4 and for pipeline 2 equaled 0.046 ± 0.003 ppm C2H6/ppm CH4. These values are in good agreement with on-site GC-FID results which reached 0.075 ppm C2H6/ppm CH4 and 0.048 ± 0.003 ppm C2H6/ppm CH4, for pipeline 1 and 2 respectively (Assan et al. 2017).

7. line 66, again, please list the measurements that were compared

A:It will be rewritten to: The results showed good agreement between the two methods (Lopez et al. 2017). Based on CRDS measurements with AirCore tool ethane to methane ratio equaled to 0.05 ± 0.01 ppm C2H6/ppm CH4, while from gas chromatography it reached 0.04 ± 0.001 ppm C2H6/ppm CH4.

8. line 88, Equation 1: rather than a generic equation, please fill in the parameters A, B, and C so that one does not have to look at the paper by Assan et al. to find these numbers

A:Below equation 1, the table with A, B, C values for low and high humidity will be added.

9. line 97. Agreed. What have the authors done to ensure comparability and traceability?

A:C2H6 was corrected for interference with H2O, CH4 and CO2 and dilution effect, using equation 1. Then, C2H6 was calibrated based on linear regression of linearity test. The scheme of these steps will be added.

10. line 105, if CMR is commonly known as precision, why not use the phrase precision?

A:The precision of a measurement can be estimated in different ways. CMR is defined specifically as "the average over 30 h of 5 min interval SD of raw data (frequency about 0.5 Hz)." (Yver-Kwok et al. 2015). CMR is an estimate of measurement uncertainty that is part of the ICOS Atmospheric thematic center protocol. We therefore decided to use CMR nomenclature for clarity and to be consistent with Yver-Kwok et al. (2015) and ICOS ATC's protocols.

11. line 153, vibrations of the instrument probably lead to instrument noise regardless of whether then instrument is "dedicated" to an ethane measurement. And are the authors referring to the ethane measurement noise when referring to "instrument readouts"? Or all measurements? And in line 154, this is referred to as a "constraint". Does this mean the mobile data were noisy to the point of being unusable?

A:By "instrument readout" we mean C2H6 concentration measured by CRDS G2201-i. The instrument noise for C2H6 and $\delta$13CH4 increases during car driving. We did not observe increased noise for CH4 mixing ratio measurements. Also, for $\delta$13CH4 we observed some additional fluctuation during crossing road bumps. Possibly, it can happen also during C2H6 measurements. Based on it, we did not use C2H6 measurements when the car was in motion and assumed it as a constraint of our approach as the uncertainty on the ratio would make it unusable

12. line 162, are these two-sided fits? Weighted by anything?

A:Fitting of the C2H6 versus CH4 was calculated as a linear regression type II (uncertainty of x- and y-axis influence fitting) with the ordinary least squares (OLS) method. Before fitting, both CH4 and C2H6 were calibrated. C2H6 was also corrected. Measured values were not weighted.

13. line 174, what is "skc"?

A:"skc flexfoil sample bag" is the product name of bags used to sample air. It will be precised in the text.

14. line 181, where has this publication been submitted? Is it available to read?

A:The publication has been submitted to Environmental Science and Technology. As the reviewing process is not public, the article is not available to read at this moment.

15. line 185, what is the purpose of this sentence? Was the change in drying intentional? if so, for what reason? Was it regular?

A:After the sentence: "Part of the measurements was made with magnesium perchlorate as a dryer before the instrument inlet and part of measurements without dryer.", the sentence: "It allowed to additionally verify the water influence on ethane to methane ratio" will be added. Later in the section 3.3, line 300, information that for humidified measurements ethane to methane ratio was higher than values provided by operator will be added.

16. line 201, the authors should define a "low" amount of ethane. It seems like they are referring to 23 ppb, which is not low. But reading later, it appears they are referring to 2.2 ppb? But that is in the next section, so I'm not sure if that is the same working standard referred to in this section. Regardless, the authors should start with their best estimate of the tank mixing ratio. This puts the G2201-i performance in perspective. Otherwise, the reader has to read several paragraphs to discover a 2.2 ppb standard reads as 23 ppb on this instrument.

A:We thank the reviewer for this comment enabling increased clarity. The order of this paragraph and materials and methods paragraph will be changed. First general laboratory set up will be explained, then interference correction and water sensitivity, followed by ethane calibration factors. In the next step CMR and Allan deviation will be described, followed by Time drift section. Different working gases were used during laboratory tests. In line 201 we presented measurements of one working gas (23

ppb of C2H6) over half a year. It is different working gas than one used during CMR and Allan Deviation (33 ppb of C2H6). The working gas, used for time drift test, was filled with dried ambient air, thus C2H6 concentration was similar to the concentration in the working gas used to measurements of CMR and Allan Deviation. In revised manuscript, used working gases will be better numbered and better described.

17. For Table 1 and Figure 1, Was this the working gas used as part of the dilution system described on page 4/equation 1? In general, I think whenever the authors mention a working gas, they should state what the nominal ethane mixing ratio is.

A:Different working gases were used for calibration (Table 1) and for time drift observation (Figure 1). The working gas used to determine calibration factors is part of the dilution system described on page 4, equation 2. Its nominal C2H6 concentration (measured by GC-FID) was equal to 2.2 ppb, while from CRDS measurement we obtained $33.2 \pm 1.7$ ppb. The second working gas, was another working gas which C2H6 concentration over 6 months was $23 \pm 12$ ppb on the CRDS. Unfortunately, during one measurement the working gas was accidentally fully released and it was not possible to measure its C2H6 concentration on the GC. The distinction between different working gases will be added/clarified in the text.

18. line 210, can you add an uncertainty to the 2.2 ppb?

A:The uncertainty will be added: $2.2 \pm 0.1$ ppb.

19. line 212, this was a surprisingly high offset. I am also still getting confused by the working gases used. This is apparently not the same one used for Figure 1? And if Figure 1 averaged 23 ppb, presumably you were giving it less than 2.2 ppb (2.2 * 23/33 = _1.5 ppb?).

A:As explained in question 17, different working gases were used for time drift (Figure 1) and Linearity test and CMR and Allan Deviation (Table 1 and Table 2). The nominal value for working gas of measured 23 ppb was unknown (question 17). However, as it

was filled with ambient air, it has a C2H6 concentration similar to another working gas, so possibly, 2.2 * 23/33 = ∼1.5 ppb.

20. line 218 and 220, there are a lot of Picarro model numbers in this paragraph. Perhaps the authors could add a table to show what models measure what species, since I am not familiar with all the models.

A:The table with instrument characteristic will be added, in the methods section, in paragraph 2.1 Laboratory set-up, before comparing different instruments.

21. Figure 2, what units are the Allan deviation plots in? I assume they are all ppm?

A:Yes, on Figure 2 Allan Deviation is presented in ppm. The axis labels will be improved in revised version.

22. line 240, in some cases such as these, a standard error of the mean would also be worth reporting, along with the standard deviation

A:The standard error will be calculated for all three protocols and their values will be added in the text. The difference between standard deviation and standard error will be also explained in the text.

23. Figure 3, are the differences between Protocol 2 and 3 simply linear fits, i.e., Protocol 2 fits a line to all the data, and Protocol 3 fits a line to data < 0.16% H2O? If so, what would a higher-order fit to H2O do – could you use that for both high and low humidity cases? I'm also not sure of the benefit of naming these "Protocol X", since every time they are mentioned, a description of the Protocol is also given. It seems easier to mention "no correction", etc. every time, and the reader wouldn't have to remember what arbitrary Protocol number this was given.

A:All 3 protocols fit all the data but protocol 1 uses no correction, protocol 2 uses the high-water content equation on all data (except the first point at 0%) and protocol 3 uses the low-water content equation on all data. The name "protocol X" will be deleted from text and it will stay with "no correction", "low humidity", "high humidity" as suggested by

the reviewer.

24. line 255, what does a release with a C2H6:CH4 equal to 0 mean? No ethane was released, but methane was? Or nothing at all was released?

A:In the case when C2H6:CH4 was equal to 0.0, yes, ethane was not released while methane was released. This information will be added in the manuscript.

25. line 264, when absolute deviations are on the order of 10 ppb, an "improvement" of 0.4 ppb seems like simple statistical variation. In other words, I think the authors are assigning significance to the insignificant digits of these numbers.

A:We agree with this comment and this part will be rewritten to show insignificant change of observed variation between raw and averaged data.

26. Table 3, Why do the authors report the residuals, and not the ratio itself? And how are the residuals defined? Is a linear fit performed on the data, and these are the residuals when the fitted line is subtracted from the data?

A:Here, residuals are presented instead of the ratio to present the difference between emitted and observed ratios. In the revised version, the table will be improved to present ratios themselves. Yes, to obtain these residuals, the fitted line is subtracted from the data. This part will be clarified in the revised manuscript.

27. Table 4, referring to the different sites as A, B, C, and D only further complicates this table. Also, I'm not yet sure what difference the survey number makes. I think it would be easier to refer to these as compressor 1, 2, 3, and landfill. Use abbreviations if necessary. Also, move the * information from the title of the table to below the table.

A:In Tables 4 and 5, numbers represent different measurements made on one site (e.g. made during different days or in different location on the site). In the revised manuscript, the terminology of sites will be clarified and the * will be moved below table.

28. Figure 5, how are these slopes calculated? Are the data weighted in the fit? And are the uncertainties reported in Table 4 just the slope uncertainties, or do they tie in the uncertainties of the C2H6 measurement?

A:As described in question 12, slopes are calculated using a linear regression type II (uncertainty of x- and y-axis influence fitting) with the ordinary least squares (OLS) method. The data are not weighted. Reported in Table 4 and Table 5 uncertainties are slope uncertainties without adding uncertainties of C2H6 measurements.

29. line 407, it might be best to reiterate the requirement that CH4 be greater than 1 ppm here, as mentioned previously in the paper

A:This part will be rewritten to highlight possibilities of using CRDS G2201-i to measure ethane to methane ratio and the requirement of 1 ppm of CH4 will be added there.

30. Grammar suggestions/typos: line 28, it looks like "sources" is possessive, needs apostrophe line 34 and elsewhere, I think "ethane:methane ratio" is redundant. Suggest either "ethane:methane" or "the ethane to methane ratio". But to me, using a colon implies ratio. line 42–43, change "methane enhancement source" to "methane source" line 45, remove "access to" line 56, change "biogenic or thermogenic" to "biogenic from thermogenic" line 85, add "1" to H in CH4 for consistency line 124, change to "Equation" line 133, change "has been measured during" to "was sampled for" line 152, change "the previous works" to "previous work" line 155, change "standing some" to "spending" line 155, change "accumulating air in" to "sampling air using" line 159, add "the" before "C2H6:CH4" lines 164–5, start sentence "A description of the experimental: : :", replace "find" with "found", and add period after "(2017)" line 167, suggest "up to" instead of "until" line 169, suggest "C2H6:CH4" instead of "ethane:methane" for consistency line 170, suggest "stationed in the plume" line 171–172, suggest ": : : the time spent within the plume was approximately 15 to 20 minutes." line 173, suggest "tracer release" line 174, change "5 liters'" to "5-liter". Also line 284. line 175, change to ": : : bags were sampled inside : : : and one was sampled : : :" line 175, change

"bags" to "bag samples" line 180, delete "real". I think "field" is sufficient. line 183, add "the" before "C2H6:CH4" lines 185–186, change "part of measurements without dryer" to either "part with a dryer" or "part of the measurements without a dryer" Figure 1 caption, I would re-word and make two sentences, change "20 minutes" to "20-minute", start new sentence with "For each measurement point, squares represent: : :" line 214, change to "As a result: : :" line 218, change "dedicated to the measure of ethane" to "designed to measure ethane" line 227, change "ethane absolute value" to "an absolute value of ethane" line 229, change "deduct" to "deduce" line 259 and elsewhere, suggest "stationary in-plume situation" instead of "plume standing situation" line 273, suggest something like: "For the higher emission, the measurements and results were combined when the emission rates were 70, 72, and 73 L/min." line 276, add "the" before "AirCore" line 285–286, add "to" after "equal" Table 4, change "Data" to "Date" line 321, change "due the very" to "due to the very" line 329, change "ratio" to "ratios" line 356–357, the time of sampling is confusing. The first sentence makes it sound like the instrument spends 10 minutes online, followed by 10 minutes offline. The next sentence makes it sound like the instrument spends 10 minutes online, followed by 20 minutes offline. line 358, perhaps just describe the CRDS data as being averaged over the sampling time of the GC-FID line 360–361, change to ": : : to use a CRDS G2201-i to measure C2H6:CH4, : : :" line 366, change to ": : : on the TILDAS method : : :" line 367, change to "tracer release" line 384–385, a word is missing here, perhaps "allowed us to", change "measurements point" to "measurement points" line 391, change to either "allow us to separate" or "allow the separation" line 394, change to "flask samples" line 398, indicative of what?

A:All suggested grammar correction and found typos by Reviewer will be corrected and after the revised manuscript will be verified again with a view to the grammar and typos. Also, the sentences in lines 356-358 will be rewritten to make it clear and consistent. The sentence in line 356 will be rewritten: "For GC-FID, ambient air was collected 10 minutes and during following 20 minutes instrument measured the input air."

for the tables in the Appendix, I would put the * asides below the table, rather than part of the table title

A:The * will be moved below table.

References Assan, Sabina, Alexia Baudic, Ali Guemri, Philippe Ciais, Valerie Gros, and Felix R. Vogel. 2017. "Characterization of Interferences to in Situ Observations of Delta13CH4 and C2H6 When Using a Cavity Ring-down Spectrometer at Industrial Sites." Atmospheric Measurement Techniques 10 (6): 2077–91. https://doi.org/10.5194/amt-10-2077-2017. Rella, C. W., J. Hoffnagle, Y. He, and S. Tajima. 2015. "Local- and Regional-Scale Measurements of CH4, △13CH4, and C2H6 in the Uintah Basin Using a Mobile Stable Isotope Analyzer." Atmospheric Measurement Techniques 8 (10): 4539–59. https://doi.org/10.5194/amt-8-4539-2015. Yver Kwok, C., O. Laurent, A. Guemri, C. Philippon, B. Wastine, C. W. Rella, C. Vuillemin, et al. 2015. "Comprehensive Laboratory and Field Testing of Cavity Ring-down Spectroscopy Analyzers Measuring H2O, CO2, CH4 and CO." Atmospheric Measurement Techniques 8 (9): 3867–92. https://doi.org/10.5194/amt-8-3867-2015.

[Figure]

---

## Author Comment (AC3) · 5 Feb 2021

Author Comments on the manuscript 10.5194/amt-2020-410-RC3, Reviewer 3

We would like to thank Reviewer 3 for the constructive comments that aided us to improve our manuscript. In this document we provide our replies to the Reviewer's comments. Following every comment, we give our reply. Here line numbers, page numbers and figure numbers refer to the original version of the manuscript.

General Comments This manuscript assesses the ethane measurement obtained using the Picarro G2201- i and tests its ability to provide meaningful data for determining
C2H6:CH4 in methane plumes, with the goal of source attribution. The instrument is tested and calibrated in the laboratory, subjected to controlled release experiments, and taken to measure real sources in the field. The authors find that, due mostly to the low precision of the ethane measurement ($\sim$ 50 ppb), the G2201-i can only realistically be used for ethane-to-methane ratios in methane peaks that are at least 1 ppm above the background. Furthermore, the measurement as presented must be taken under stationary conditions (i.e., with the mobile platform parked within a plume for $\sim$30 min) or the noise of the ethane measurement becomes unacceptably high. The use of the G2201-i for the described applications seems extremely limited, especially in light of the other available instruments that can do this type of measurement much better (LGR, Aerodyne, and other Picarro models). However, the authors do recognize that in order to use the Picarro G2201-i for ethane field measurements (which in turn are to be used only in the calculation of ethane-to-methane ratios rather than absolute ethane mixing ratios), the instrument response must be extensively characterized. This work is done, and the limitations of the G2201-i for the purposes described are appropriately determined and discussed.

1. There is a lot of information presented on the experimental details of previous work, which, in my opinion, obscures the experimental design and the main conclusions of the current manuscript somewhat. It makes it difficult for the reader to focus on the important points of the manuscript (one of which is the many conditions that need to be satisfied to obtain useful ethane information from the G2201-i). I recommend the authors try to streamline the manuscript as much as possible so that the important points are evident. Additionally, I recommend careful proofreading of the manuscript, which contains many small grammar errors, some of which are highlighted below under "Technical Comments".

A:The manuscript will be "streamlined" as suggested to present the study more clearly in order to be useful for other scientific teams which already have CRDS G2201-i and would like to use it in field conditions for measuring both $\delta$13CH4 and ethane

[Figure]

to methane ratio. Thus, we will modify our manuscript for that. It can be treated as a protocol where all necessary steps are described and verified before field work. To do it, in the revised manuscript the introduction and conclusion parts will be improved to highlight the importance of the work done. Also, the method section will be rewritten to make it more straightforward and some additional explanation will be added to make it more "stand-alone" work (e.g adding scheme or conducted test before using instrument on field or adding table to equation 1, with factors A, B, C for different humidity levels).

Specific Comments 2. Lines 53-58: How does this study differ from Assan et al? Is the system just characterized better? Is the only difference, as mentioned later in lines 361+, that the instrument was put in a car (which must remain stationary within a plume for ∼30 mins to take a useful measurement)? If so, that should be made clear early on.

A:In our study, additional tests were made, the previously calculated correction and calibration factors were evaluated and long term drift was verified. Notably, we did not observe the time drift, contrary to Assan et al. Also, compared to Assan et al., a controlled release experiment was made. Ultimately, we wanted to check in which conditions we can measure ethane to methane ratio in short time, near-source conditions. Overall, we showed it is possible to received reliable values during short time (i.g. 30 minutes) and the instrument can be installed inside the car. Having the instrument set-up inside the car facilitates the measurement set-up as an additional place to install the stationary instrument is not required anymore. However yes the measurements are field-tested with the car idling. This explanation will be added in the method section.

3. Lines 62+: Did you use the monitoring mode in addition to the replay mode for the Aircore in the current study? I think some more information on how the Aircore was used specifically for this study should be included, although I would add this information later in the methods section.

A:Indeed, all measurements which were made during standing inside the plume were made in the monitoring mode. Also, we drived through CH4 plumes (in monitoring mode) and remeasured air accumulated in the AirCore sampler (replay mode). However, during car motion, the instrument noise increased and also crossing road bumps can cause additional fluctuation of measured C2H6. Thus, comparison of data collected in monitoring mode and replay mode, where the same plume is remeasured, can by biased due to influence of car motion for C2H6 readout in monitoring mode. This information will be added in the method section.

4. Lines 82 – 85: This background information on how ethane is measured and reported for an isotopic methane/carbon dioxide instrument should be moved to the abstract and introduction.

A:This information will be moved to the introduction section.

5. Lines 81+ (Materials and Methods section): To make each factor investigated clear, consider re-formatting with subheadings, such as, 1.1 Laboratory 1.1.1 Interference Correction on Ethane and Water Sensitivity 1.1.2 Ethane Calibration Factors 1.1.3 Precision and Allan Variance 1.1.4 Time drift Because the water vapor sensitivity tests are tests on the validity of the interference corrections, I think this should be discussed at the same time as the interference correction in general.

A:Material and Method section will be rewritten according to this comment. These changes will be followed by changed order in the results paragraph.

6. Lines 147+: I have some confusion about what Protocol 1, 2, and 3 are. Are these described clearly somewhere? I would add relevant details here in the methods section.

A:Protocols 1,2,3 are arbitrary made protocols to describe which correction factors were used. Thus Protocol 1 is when no interference correction was used, Protocol 2 when interference correction was used for high humidity case and Protocol 3 for low

humidity case. However, as Protocol X is always followed by a short description of cases, in the revised manuscript we will remove "Protocol X" and will mention for which case results are presented (i.g. "no correction", "high humidity" or "low humidity").

7. Lines 151-152: Delete "The measurement setup used here is the same as in the field" and only mention in section 2.3.

A:This correction will be applied.

8. Lines 154-159: The point that true "mobile" measurements are not conducted (i.e., while the vehicle is moving) should be highlighted earlier in the manuscript. It is an important point that is somewhat hidden here. Also- please add information here about the specifics of the Aircore setup as used in this study (e.g., flow rates, different modes, car stopped or moving).

A:This information will be added in the method section (question 3). Also the details about how to use AirCores will be added.

9. Line 174- 178: I question whether any of the information about the failed bag measurements should be included in the main manuscript, especially given the issues with sampling and bag preparation mentioned later (in lines 284+). Maybe make a very abbreviated reference to them, and then move all other bag information to the supporting information.

A:During preparing the manuscript, we considered the same question. In the revised manuscript, we will move this bag samples part to Appendix and leave only a short explanation in the main text.

10. Lines 195+, Section 3.1: Suggest headings that are the same as those suggested above for section 2.1 to help organize the information.

A:The heading will be changed and text will be rewritten according to suggestion in question 5.

11. Line 201: Can you specify what a "low amount of C2H6" means?

A:This working gas was fulfilled with dried ambient air, thus C2H6 concentration was similar to concentration of 2.2 ppb in working gas used to measurements of CMR and Allan Deviation. In the revised manuscript, used working gases will be better numbered and better described.

12. Line 356: Please clarify what "10 mins of ambient air collection was measured during 20 minutes" means.

A:Sentences in lines 356-358 will be rewritten to make it clean and consistent. The sentence in line 356 will be rewritten: "For GC-FID, ambient air was collected 10 minutes and during following 20 minutes the instrument analysed the input air."

13. Lines 384-390: Please revise this section on Aircore and CRDS flow rates for clarity. How are the Aircore and CRDS flows related? Were there reasons for the chosen flows?

A:In the set-up used in this study, the CRDS had flow 160 mL/min and this flow rate was used in the monitoring mode. Then in the replay mode, using the needle valves, the flow rate decreased to about 50 mL/min. It caused the three times increase of the number of measurement points. In our set-up, the instrument flow rate in monitoring mode was increased (by default, in CRDS G2201-i the flow rate is equal to 25 mL/min) to achieve faster instrument response during mobile measurements. Then, the flow rate in the reply mode was chosen as optimal solution between increasing the number of measurement points and having enough air for each zone sampled. With 50 mL/min flow rate, one AirCore analysis lasts about ten minutes. Smaller flow rate would allow to increase the number of measurement points and thus instrument precision, but also requires longer time to use replay mode and measure air stored in AirCore. The Air-Core tool was also used inside Paris city and we decided to keep 10 minutes sampling time of replay mode. As Paris is a crowded city with numerous traffics, it would not be possible to stop the car for longer analysing time to measure CH4 plume in the replay

mode. The additional explanation of AirCore and CRDS flow rates will be added.

14. Line 398: Do you mean the "first comparisons" of ethane mixing ratios with GC-FID match up in a relative sense? The word "indicative" is confusing here.

A:In this line we mean first comparison between CRDS sampling according to the protocol presented in the manuscript with GC-FID. The sentence will be rewritten for clarity.

15. Line 431: Please clarify what the "flushing issue" to be solved is.

A:Here, by flushing issue we mean a too small decrease of the flow rate. A stronger decrease of the flow rate will result in an increase of measurements points and an improvement of instrument accuracy. Further decreasing flow rate could solve the problem of observed differences between AirCore and actual ethane to methane ratio (discussion in lines 382- 390). In line 431 it will be rephrased for clarity.

16. Technical Comments Line 43: "source" should be "sources" Line 54: "measure of" should be "measurement of" Line 60: Change "allows to improve time resolution" to "allows improvement of time resolution" Line 77: Change "instrument to ethane" to "instrument for ethane" Line 165: Change "find" to "found" Line 167: Change "emission flux" to "gas flow rate" Line 168: Change "could vary" to "were varied"

A:All suggested grammar corrections and found typos by Reviewer will be corrected and after the revised manuscript will be verified again with a view to the grammar and typos.

---

## Author Response (AR1)

**Author Comments on the manuscript 10.5194/amt-2020-410-RC1, Reviewer 1**

We would like to thank Reviewer 1 for his comments that helped us to improve our manuscript. We provide below a detailed reply to the Reviewer's comment on the utility of our work. Manuscript will be clarified accordingly. The original comments made by the Reviewer are typeset in italic and bold face font. The authors' changes in the manuscript are typeset in italic face font and line number refer to the new version of the manuscript.

***The stated main objective of this paper: is to evaluate the performance of the CRDS2201-i and the applicability of making short-term, direct, continuous, mobile measurements of ethane in methane-enriched air, with sufficient precision during near source ("pollution plume conditions") surveys. The authors did a commendable amount of work characterizing their instrument and in presenting all the limitations of this instrument. This is to their credit, and to this end the work described herein achieves its objectives.***
***However, with that being said, this reviewer finds very limited applications where the G2201-i analyzer can be employed in measuring ethane/methane slopes in real world situations. As stated, peak ethane enhanced values of at least 100 ppb and peak methane values of at least 1 ppm are needed on stationary platforms for meaningful slopes. Unless one is directly at a well head or at a compressor source, this type of performance is not very useful. Also, more discussions of the 30 ppb bias in their ethane measurements, its sources, and its variability are needed.***
***Despite the body of work here, this reviewer does not find any utility in publishing this paper with very limited real world applications for the G2201-i analyzer in terms of ethane/methane slopes. This reviewer recommends that the authors instead focus on a similar concerted effort to characterize their CRDS 2210-i analyzer, which they briefly mention, and shows superior performance for ethane.***

We think that the full characterization of **CRDS 2201-i analyzer** to measure ethane to methane ratios proposed in our paper is useful and worth publishing for the following reasons:

1.      **Valuable opportunities exist for using this instrument beyond its intended application (i.e. measuring ethane together with isotopic composition on a single analyzer), but this requires a prior specific characterization**. In our study we focused on the characterization of CRDS G2201-i for ethane measurements, as some previous studies already used this instrument during field campaigns to measure ethane to methane ratio in fixed settings such as a shelter (described in paragraph 4 discussion). Thus our purpose was to evaluate limitation and possibilities to use this instrument to measure ethane to methane ratio in a car setting (one conclusion was that, indeed, it needs to be stationary during measurements but is mobile over a site). This study is useful for other scientific teams, which do not have an instrument dedicated for ethane measurements, but already have the CRDS G2201-i and would like to use it in field conditions for measuring both $\delta^{13}CH_4$ and ethane to methane ratio. According to our knowledge, outside our team, two other research teams in Europe use CRDS G2201-i during mobile measurements (Heidelberg University and AGH University of Science and Technology). Possibly, more institutions use it as well. Thus, our manuscript can be viewed as a protocol where all necessary

steps are described and verified before field work. On a side note, the CRDS 2210-i briefly discussed was only tested in our colleagues' laboratory but was on loan from another institute. It was not available for us to test in field conditions.

[revised manuscript text omitted]

**References**

Assan, Sabina, Alexia Baudic, Ali Guemri, Philippe Ciais, Valerie Gros, and Felix R. Vogel. 2017. “Characterization of Interferences to in Situ Observations of Delta13CH4 and C2H6 When Using a Cavity Ring-down Spectrometer at Industrial Sites.” *Atmospheric Measurement Techniques* 10 (6): 2077–91. https://doi.org/10.5194/amt-10-2077-2017.

Hoheisel, Antje, Christiane Yeman, Florian Dinger, Henrik Eckhardt, and Martina Schmidt. 2019. “An Improved Method for Mobile Characterisation of Δ13CH4 Source Signatures and Its Application in Germany.” *Atmospheric Measurement Techniques* 12 (2): 1123–39. https://doi.org/10.5194/amt-12-1123-2019.

Lan, Xin, Pieter Tans, Colm Sweeney, Arlyn Andrews, Edward Dlugokencky, Stefan Schwietzke, Jonathan Kofler, et al. 2019. “Long-Term Measurements Show Little Evidence for Large Increases in Total U.S. Methane Emissions Over the Past Decade.” *Geophysical Research Letters* 46 (9): 4991–99. https://doi.org/10.1029/2018GL081731.

Lopez, M., O.A. Sherwood, E.J. Dlugokencky, R. Kessler, L. Giroux, and D.E.J. Worthy. 2017. “Isotopic Signatures of Anthropogenic CH4 Sources in Alberta, Canada.” *Atmospheric Environment* 164 (September): 280–88. https://doi.org/10.1016/j.atmosenv.2017.06.021.

Lowry, David, Rebecca E. Fisher, James L. France, Max Coleman, Mathias Lanoisellé, Giulia Zazzeri, Euan G. Nisbet, et al. 2020. “Environmental Baseline Monitoring for Shale Gas Development in the UK: Identification and Geochemical Characterisation of Local Source Emissions of Methane to Atmosphere.” *Science of The Total Environment* 708 (March): 134600. https://doi.org/10.1016/j.scitotenv.2019.134600.

Nisbet, E. G., M. R. Manning, E. J. Dlugokencky, R. E. Fisher, D. Lowry, S. E. Michel, C. Lund Myhre, et al. 2019. “Very Strong Atmospheric Methane Growth in the 4 Years 2014–2017: Implications for the Paris Agreement.” *Global Biogeochemical Cycles* 33 (3): 318–42. https://doi.org/10.1029/2018GB006009.

Saunois, Marielle, Ann R. Stavert, Ben Poulter, Philippe Bousquet, Joseph G. Canadell, Robert B. Jackson, Peter A. Raymond, et al. 2020. “The Global Methane Budget 2000–2017.” Preprint. Atmosphere – Atmospheric Chemistry and Physics. https://doi.org/10.5194/essd-2019-128.

Schwietzke, Stefan, Owen A. Sherwood, Lori M. P. Bruhwiler, John B. Miller, Giuseppe Etiope, Edward J. Dlugokencky, Sylvia Englund Michel, et al. 2016. “Upward Revision of Global Fossil Fuel Methane Emissions Based on Isotope Database.” *Nature* 538 (7623): 88–91. https://doi.org/10.1038/nature19797.

Sherwood, Owen A., Stefan Schwietzke, Victoria A. Arling, and Giuseppe Etiope. 2017. “Global Inventory of Gas Geochemistry Data from Fossil Fuel, Microbial and Burning Sources, Version 2017.” *Earth System Science Data* 9 (2): 639–56. https://doi.org/10.5194/essd-9-639-2017.

Turner, Alexander J., C. Frankenberg, and Eric A. Kort. 2019. "Interpreting Contemporary Trends in Atmospheric Methane." *Proceedings of the National Academy of Sciences* 116 (8): 2805–13. https://doi.org/10.1073/pnas.1814297116.

Yacovitch, Tara I., Conner Daube, and Scott C. Herndon. 2020. "Methane Emissions from Offshore Oil and Gas Platforms in the Gulf of Mexico." *Environmental Science & Technology* 54 (6): 3530–38. https://doi.org/10.1021/acs.est.9b07148.

Yacovitch, Tara I., Scott C. Herndon, Joseph R. Roscioli, Cody Floerchinger, Ryan M. McGovern, Michael Agnese, Gabrielle Pétron, et al. 2014. "Demonstration of an Ethane Spectrometer for Methane Source Identification." *Environmental Science & Technology* 48 (14): 8028–34. https://doi.org/10.1021/es501475q.

Zazzeri, G., D. Lowry, R.E. Fisher, J.L. France, M. Lanoisellé, and E.G. Nisbet. 2015. "Plume Mapping and Isotopic Characterisation of Anthropogenic Methane Sources." *Atmospheric Environment* 110 (June): 151–62. https://doi.org/10.1016/j.atmosenv.2015.03.029.

**Author comments on the manuscript 10.5194/amt-2020-410-RC2, Reviewer 2**

We would like to thank Reviewer 2 for the constructive comments that aided us to improve our manuscript. In this document we provide our replies to the Reviewer's comments. The original comments made by the Reviewer are typeset in italic and bold face font. Following every comment, we give our reply. Here line numbers, page numbers and figure numbers refer to the original version of the manuscript. The authors' changes in the manuscript are typeset in italic face font and line number refer to the new version of the manuscript.

*Defratyka et al. report ethane measurements from a spectroscopic instrument originally not designed to make an ethane measurement. Ethane has a small interfering absorption peak for an instrument that reports isotopic measurements of $CO_2$ and $CH_4$. Defratyka et al. quantify this peak, and although the ethane measurement has low precision, use it to quantify the ethane to $CH_4$ ratio from natural gas emissions. Although the ethane measurement is not very good, and the application of this measurement is limited, it nonetheless could be of some use to the scientific community.*
*However, before this paper is ready for publication, I think some issues must be addressed.*

*1. The Picarro 2201 website (https://www.picarro.com/sites/default/files/product_documents/Picarro_%20G2201-i%20Analyzer%20Datasheet_053017.pdf) says there are interferences from "other organics", as well as ethane, ammonia, ethylene, and sulfur-containing compounds. Might some of these other organic exist in natural gas? Have the authors looked at propane interferences?*
Rella et all (2015) quantified the influence of other organic compounds for $\delta^{13}CH_4$ using CRDS G2132, which operates in the same wavelengths as CRDS G2201-i. They also noted that ammonia was having a strong influence on ethane. No other organic compounds from Table 1 tested in their paper were noted as having an influence. As CRDS G2132 and CRDS G2201-i operate in the same wavelength, the observed interferences are similar for both instruments.
CRDS G2201-i has the possibility to measure $H_2S$, $NH_3$ and $C_2H_4$. Similarly, to $C_2H_6$ measurements, they are measured to account for their interference for $\delta^{13}CH_4$ and, similarly to $C_2H_6$ measurements, they should be calibrated and corrected before any use and large instrument noise is observed during their measurements. During our study, no signal above instrument noise was observed for $H_2S$, $NH_3$ and $C_2H_4$ so we neglected their interference. Unfortunately, with CRDS G2201-i, it is not possible to measure $C_3H_8$, so we cannot conclude about possible propane interference from our measures. However, as said before, no interference on ethane was noted for propane in Rella et al. Thus, we assume that propane interference is negligible.

line 96 "*Interferences with other species is presented in Appendix A.*"
lines 525-534 "*Rella et all (2015) quantified the influence of other organic compounds for $\delta^{13}CH_4$ using CRDS G2132, which operates in the same wavelengths as CRDS G2201-i. They also noted that ammonia was having a strong influence on ethane. No other compounds from Table 1 (e.g. CO, $CH_3SH$) tested in their paper were noted as having an influence. As CRDS G2132 and CRDS G2201-i operate in the same wavelength, the observed interferences are similar for both*

*instruments. CRDS G2201-i has the possibility to measure $H_2S$, $NH_3$ and $C_2H_4$. Similarly, to $C_2H_6$ measurements, they are measured to account for their interference for $\delta^{13}CH_4$ and, similarly to $C_2H_6$ measurements, they should be calibrated and corrected before any use and large instrument noise is observed during their measurements. During our study, no signal above instrument noise was observed for $H_2S$, $NH_3$ and $C_2H_4$ so we neglected their interference. Unfortunately, with CRDS G2201-i, it is not possible to measure $C_3H_8$, so we cannot conclude about possible propane interference from our measures. However, as said before, no interference on ethane was noted for propane in Rella et al (2015). Thus, we assume that propane interference is negligible."*

**2.      I was confused in the second sentence of the Abstract by the use of the word "dedicated", which is also used throughout the paper. To me, "dedicated" means it only measures that to which it is dedicated, in this case $CO_2$ and $CH_4$. I suggest the authors use "originally designed to measure" or some such phrase. And it wasn't until line 84 on page 3 that the authors mention for the first time that the G2201-i was actually used to measure ethane. The authors should explicitly state that the G2201-i was used to measure ethane in the Abstract, rather than hiding it in terms of "consider[ing] the possibility" of measuring ethane.**

In the revised version, "Dedicated" will be changed into "originally designed to measure" in abstract and in the rest of the manuscript. Also, it will be clearly said in the abstract and introduction that CRDS G2201-i was used to measure $C_2H_6$.

*lines 13-17 "In our study, we characterized the CRDS Picarro G2201-i instrument, originally designed to measure isotopic $CH_4$ and $CO_2$, for measurements of ethane to methane ratio in mobile, near-sources, field conditions. We evaluated the limitations and potential of using the CRDS G2201-i to measure ethane to methane ratio, thus extending the instrument application to measure simultaneously two methane sources proxies in the field: carbon isotopic ratio and ethane to methane ratio."*

*lines 60-63 "Here, building on previous studies with CRDS instruments, we detail the possibilities and limitations of measuring $C_2H_6$ using the CRDS G2201-i, in the vicinity of methane source. The CRDS G2201-i is originally designed to measure $^{12}CO_2$, $^{13}CO_2$, $^{12}CH_4$, $^{13}CH_4$ and $H_2O$ and record $C_2H_6$ only as an internal way to correct $^{13}CH_4$, thus observed $C_2H_6$ mixing ratio must be corrected and calibrated."*

*lines 96-99 "By default, $C_2H_6$ is not intended for use by standard users. Thus, the measured $C_2H_6$ mixing ratio is not corrected nor calibrated and it is stored in private archived files. To use ethane measurements per se, measured $C_2H_6$ values must be first corrected for interferences with $^{12}C^{16}O_2$ (further $CO_2$), $H_2O$ and $^{12}CH_4$."*

**3.      Generally, I thought the paper needed more statements of introduction and conclusion in many paragraphs. There are a lot of paragraphs explaining what the authors did related to the measurement. What is missing is information on why they are doing this, and what are the results of this part of the experiment.**

We will improve the introduction to highlight importance of mobile, near source mobile measurements. Also results and conclusions of every part of experiment will be detailed and explained. Overall application and significance of work will be described.

*lines 76-86 "Here, the main purpose of this study is to evaluate the performances of the CRDS G2201-i and the applicability of making short-term, direct, continuous, mobile measurements of ethane in methane-enriched air, with sufficient precision during near-source surveys. Our motivation is to perform both isotopic and ethane measurements with only one instrument in the field in order to improve the partition of methane sources without the need for an additional analyzer. We aim at providing a protocol useful for other scientific teams, that do not have an analyzer designed for ethane measurements, but already have the CRDS G2201-i and intend to use it in field conditions for measuring both $\delta^{13}CH_4$ and ethane to methane ratio.*

*To achieve this goal, the first step consists of laboratory tests to calculate the calibration factors and also to check the instrument performances in stationary, laboratory conditions extending preliminary work by Assan et al. (2017). The second, novel step evaluates the performances of the instrument during mobile field measurements in a controlled release experiment. A mixture with known $C_2H_6:CH_4$ and $CH_4$ emission flux was released and compared to measured ratios from CRDS G2201-i and LGR UMEA."*

*lines 351-353 "During the release experiment, we showed that the CRDS is able to separate the different emitted mix through their $C_2H_6:CH_4$. Standing in the plume resulted in a better agreement with the real ratios, with less spread of the residuals than using AirCore sampling. Increasing the AirCore sampling frequency could potentially help resolve this discrepancy."*

*lines 407-409 "Field work allowed us to compare our measurements against operator values and GC measurements. This confirms that this instrument can discriminate between sources and that it agrees within its uncertainty with more precise methods such as GC."*

*lines 444-447 "Even though, we showed it is possible to receive reliable values during short time (e.g. 35 minutes) and the instrument can be installed inside a car. Notably, having the instrument setup inside the car facilitates the measurement setup, as an additional place to install the stationary instrument is not required anymore."*

*lines 486-488 "Also, this instrument can be used to observe possibly temporal variation of $C_2H_6:CH_4$ of methane emitted from fossil fuel sources. These studies can be made in the vicinity of strong emitting sources, where $CH_4$ plume reaches at least 1 ppm above background."*

**4.    I also would like to see this paper act more as a stand-alone work. As written, it is tied heavily to Assan et al. (2017) in too many places. In many cases, a sentence or two summarizing the results of the cited work would be helpful.**

The method section will be rewritten to make it clearer and complete, to be self-independent. Below equation 1, table with values of factors A, B, C for different water vapor levels will be added. Also, a scheme of necessary steps to calibrated and correct $C_2H_6$ will be added. The set-up of linearity test will be also added.

*lines 106-107 "Figure 1 Flow chart of steps to use $C_2H_6$ measured by CRDS G2201-i. The number in the corner corresponds to the subsection where methods of each step are presented."*

*lines 118-124 "The cross sensitivities with $H_2O$, $CO_2$ and $^{12}CH_4$ induce a bias in raw $C_2H_6$ observed by CRDS G2201-i. Assan et al. (2017) provided the strategy to determine $C_2H_6$ correction factors to account for these interferences. During the experiment, the $C_2H_6$ mixing ratio of measured gas mixture was constant, while the mixing ratio of interfering species was changed and controlled using a setup similar to the one presented on Fig. 2 in the Sect. 2.1.2. During one measurement set, the concentration of only one interfering species was changing, while the concentration of other species stayed stable. The measurement set was repeated while varying concentrations of $H_2O$, $CH_4$ and $CO_2$ were adjusted. Using linear regression, the test yielded values for the interference correction factors A, B, C in Eq. 1:"*

*line 135 Table 2 Interference correction on $C_2H_6$ (Assan et al., 2017)*

*line 164 Figure 2. Experimental setup used during laboratory tests.*

*lines 306-307 "Moreover, during studies of Assan et al. (2017), bigger changes in determined calibration factors were observed over time (i. g. 60 ppb difference of factor E)."*

**Other comments:**

**5.       line 36, somewhere it should be stated that the ratios referred to in the paper are molar ratios, as opposed to mass ratios**

The sentence will be added: Based on mobile measurements, as $CH_4$ and $C_2H_6$ mixing ratios are measured, ethane to methane ratio is calculated as molar ratio.

*lines 40-41 "The reported ratios (calculated as molar ratio when based on atmospheric measurements) depend on the type of facilities and type of the reservoirs:"*

**6.       line 58, instead of simply stating "good agreement", add what measurements agreed well in case the reader is not familiar with Assan et al. (2017)**

The sentence will be added: Ethane:methane ratio from flask samples allowed to distinguish methane emissions from the two pipelines. The natural gas in pipeline 1 had ratio equaled to 0.074 $\pm$ 0.001 ppm $C_2H_6$/ppm $CH_4$ and for pipeline 2 equaled 0.046 $\pm$ 0.003 ppm $C_2H_6$/ppm $CH_4$. These values are in good agreement with on-site GC-FID results which reached 0.075 ppm $C_2H_6$/ppm $CH_4$ and 0.048 $\pm$ 0.003 ppm $C_2H_6$/ppm $CH_4$, for pipeline 1 and 2 respectively (Assan et al. 2017).

*The sentence was moved from the introduction to the discussion and improved, lines 433-438 "Moreover, during that study, flask samples were collected and further analyzed in the laboratory. $C_2H_6$:$CH_4$ from flask samples allowed to distinguish methane emissions from the two pipelines. The natural gas in pipeline 1 had an ethane to methane ratio equal to 0.074 $\pm$ 0.001 and for pipeline 2 equal to 0.046 $\pm$ 0.003. These values are in good agreement with on-site GC-FID results which reached 0.075 and 0.048 $\pm$ 0.003, for pipeline 1 and 2 respectively (Assan et al., 2017). Thus, the laboratory values showed good agreement between field, installed in the shelter, CRDS G2201-i and GC-FID results (Assan et al. 2017)."*

**7.      line 66, again, please list the measurements that were compared**

It will be rewritten to: The results showed good agreement between the two methods (Lopez et al. 2017). Based on CRDS measurements with AirCore tool ethane to methane ratio equaled to 0.05 ± 0.01 ppm $C_2H_6$/ppm $CH_4$, while from gas chromatography it reached 0.04 ± 0.001 ppm $C_2H_6$/ppm $CH_4$.

*The sentence was moved from the introduction to the discussion and improved, lines 461-462 "Based on CRDS measurements with AirCore sampler, ethane to methane ratio equaled to 0.05 ± 0.01, while from gas chromatography it reached 0.04 ± 0.001."*

**8.      line 88, Equation 1: rather than a generic equation, please fill in the parameters A, B, and C so that one does not have to look at the paper by Assan et al. to find these numbers**

Below equation 1, the table with A, B, C values for low and high humidity will be added.

*line 135 Table 2 Interference correction on $C_2H_6$ (Assan et al., 2017)*

**9.      line 97. Agreed. What have the authors done to ensure comparability and traceability?**

$C_2H_6$ was corrected for interference with $H_2O$, $CH_4$ and $CO_2$ and dilution effect, using equation 1. Then, $C_2H_6$ was calibrated based on linear regression of linearity test. The scheme of these steps will be added.

*lines 96-101 "By default, $C_2H_6$ is not intended for use by standard users. Thus, the measured $C_2H_6$ mixing ratio is not corrected nor calibrated and it is stored in private archived files. To use ethane measurements per se, measured $C_2H_6$ values must be first corrected for interferences with $^{12}C^{16}O_2$ (further $CO_2$), $H_2O$ and $^{12}CH_4$. Different interference correction factors are needed in the absence or presence of water vapor (Assan et al. 2017). These correction factors are used and discussed in light of our new tests in Sect. 2.1.1. The water sensitivity test is also described in Sect. 2.1.1. To ensure comparability and traceability of the ethane measurement, ethane measured by the G2201-i must eventually be linked to a widely used scale. Therefore, ethane values were calibrated before use (Sect. 2.1.2)."*

**10.      line 105, if CMR is commonly known as precision, why not use the phrase precision?**

The precision of a measurement can be estimated in different ways. CMR is defined specifically as "the average over 30 h of 5 min interval SD of raw data (frequency about 0.5 Hz)." (Yver-Kwok et al. 2015). CMR is an estimate of measurement uncertainty that is part of the ICOS Atmospheric thematic center protocol. We therefore decided to use CMR nomenclature for clarity and to be consistent with Yver-Kwok et al. (2015) and ICOS ATC's protocols.

*lines 109-110 "Additionally, as a part of laboratory tests, continuous measurement repeatability (CMR, used as a precision in Yver Kwok et al., 2015)"*

**11.      line 153, vibrations of the instrument probably lead to instrument noise regardless of whether then instrument is "dedicated" to an ethane measurement. And are the authors**

*referring to the ethane measurement noise when referring to "instrument readouts"? Or all measurements? And in line 154, this is referred to as a "constraint". Does this mean the mobile data were noisy to the point of being unusable?*

By "instrument readout" we mean $C_2H_6$ concentration measured by CRDS G2201-i. The instrument noise for $C_2H_6$ and $\delta^{13}CH_4$ increases during car driving. We did not observe increased noise for $CH_4$ mixing ratio measurements. Also, for $\delta^{13}CH_4$ we observed some additional fluctuation during crossing road bumps. Possibly, it can happen also during $C_2H_6$ measurements. Based on it, we did not use $C_2H_6$ measurements when the car was in motion and assumed it as a constraint of our approach as the uncertainty on the ratio would make it unusable

*line 188-189 "As the analyzer is not originally designed for mobile measurements, the vibrations induced by car motion cause noise in the instrument readouts of $C_2H_6$ mixing ratio."*

*Also the explanation of AirCore sampler is added, lines 195-202 "Here, both stopping inside the plume and AirCore approaches were used during mobile measurements. The AirCore used in this study is made of 50 m Decabon storage tube. In our setup, the instrument flow rate in the monitoring mode was increased to 160 mL min$^{-1}$ (by default, in CRDS G2201-i the flow rate is equal to 25 mL min$^{-1}$) to achieve faster instrument response during mobile measurements. Then, the flow rate in the reply mode was chosen as the optimal solution between increasing the number of measurement points and having enough air for each zone sampled. Here, in the replay mode, using the needle valves, the flow rate decreased about 3 times. With 50 mL min$^{-1}$ flow rate, one AirCore analysis lasts about ten minutes. In the replay mode, the car was stopped to avoid possible increase of instrumental noise due to car vibration. While stopping inside the plume, the data were collected in the monitoring mode with engine stopped."*

**12.      line 162, are these two-sided fits? Weighted by anything?**

Fitting of the $C_2H_6$ versus $CH_4$ was calculated as a linear regression type II (uncertainty of x- and y-axis influence fitting) with the ordinary least squares (OLS) method. Before fitting, both $CH_4$ and $C_2H_6$ were calibrated. $C_2H_6$ was also corrected. Measured values were not weighted.

*lines 205-207 "Fitting of the $C_2H_6$ versus $CH_4$ was calculated as a linear regression type II (uncertainty of x- and y-axis influence fitting) with the ordinary least squares (OLS) method. Before fitting, both $CH_4$ and $C_2H_6$ were calibrated. $C_2H_6$ was also corrected (Fig. 1)."*

**13.      line 174, what is "skc"?**

"skc flexfoil sample bag" is the product name of bags used to sample air. It will be precised in the text.

*line 222 "Three other releases were measured using sampling 5-liter bags (Flexfoil, SKC Inc.) only."*

**14.	line 181, where has this publication been submitted? Is it available to read?**

The publication has been submitted to Environmental Science and Technology. As the reviewing process is not public, the article is not available to read at this moment.

*Eventually, this citation was deleted.*

**15.	line 185, what is the purpose of this sentence? Was the change in drying intentional? if so, for what reason? Was it regular?**

After the sentence: "Part of the measurements was made with magnesium perchlorate as a dryer before the instrument inlet and part of measurements without dryer.", the sentence: "It allowed to additionally verify the water influence on ethane to methane ratio" will be added. Later in the section 3.3, line 300, information that for humidified measurements ethane to methane ratio was higher than values provided by operator will be added.

*lines 233-234 "It allowed to additionally verify the water influence on ethane to methane ratio observed by CRDS G2201-i."*

**16.	line 201, the authors should define a "low" amount of ethane. It seems like they are referring to 23 ppb, which is not low. But reading later, it appears they are referring to 2.2 ppb? But that is in the next section, so I'm not sure if that is the same working standard referred to in this section. Regardless, the authors should start with their best estimate of the tank mixing ratio. This puts the G2201-i performance in perspective. Otherwise, the reader has to read several paragraphs to discover a 2.2 ppb standard reads as 23 ppb on this instrument.**

We thank the reviewer for this comment enabling increased clarity. The order of this paragraph and materials and methods paragraph will be changed. First general laboratory set up will be explained, then interference correction and water sensitivity, followed by ethane calibration factors. In the next step CMR and Allan deviation will be described, followed by Time drift section.

Different working gases were used during laboratory tests. In line 201 we presented measurements of one working gas (23 ppb of $C_2H_6$) over half a year. It is different working gas than one used during CMR and Allan Deviation (33 ppb of $C_2H_6$). The working gas, used for time drift test, was filled with dried ambient air, thus $C_2H_6$ concentration was similar to the concentration in the working gas used to measurements of CMR and Allan Deviation. In revised manuscript, used working gases will be better numbered and better described.

*In the revised manuscript, for the laboratory tests, the order was changed as follows, both for Material and Methods section and Result section:*
*2.1. Laboratory setup*
*    2.1.1. Sensitivity of interference correction parameters to humidity*
*    2.1.2. Ethane Calibration Factors*
*    2.1.3. Precision and Allan Variance*
*    2.1.4 Time drift*

*Also, all target gasses are numbered (Target Gas 1-5).*

*Line 302-303 "Figure 5 shows the time series of Target Gas 5 measurements with an ambient amount of $C_2H_6$ during the period of December 2018-May 2019."*

*Lines 310-312 "It should be noted that the $C_2H_6$ concentration of Target Gas 5 was in the range of clean continental air (0.5-2 ppb). The observed mean $C_2H_6$ mixing ratio for Target Gas 5, equal to 23 ppb, is overestimated. This is comparable to the 31 ppb bias observed during 24 hours measurements of Target Gas 3 (Sect. 3.1.3)."*

**17.     For Table 1 and Figure 1, Was this the working gas used as part of the dilution system described on page 4/equation 1? In general, I think whenever the authors mention a working gas, they should state what the nominal ethane mixing ratio is.**

Different working gases were used for calibration (Table 1) and for time drift observation (Figure 1). The working gas used to determine calibration factors is part of the dilution system described on page 4, equation 2. Its nominal $C_2H_6$ concentration (measured by GC-FID) was equal to 2.2 ppb, while from CRDS measurement we obtained $33.2 \pm 1.7$ ppb. The second working gas, was another working gas which $C_2H_6$ concentration over 6 months was $23 \pm 12$ ppb on the CRDS. Unfortunately, during one measurement the working gas was accidentally fully released and it was not possible to measure its $C_2H_6$ concentration on the GC. The distinction between different working gases will be added/clarified in the text.

*Target Gas used for ethane calibration is described in Sect. 2.1.2, lines 153-154 "The calibration factors are calculated with the $C_2H_6$:$CH_4$ gradually increased from 0.00 to 0.15 and measured in steps of 20 minutes. This measurement cycle is repeated three times. The used target gas has an ethane mixing ratio ~52 ppm (hereafter referred to as Target Gas 2) and is mixed with the dilution gas via two MFCs (Fig. 2)."*

*Target Gas used for observation of time drift is described in Sect 2.1.4, lines 176-179 "A known working gas (dry atmospheric mixing ratio of $CH_4$ and $C_2H_6$), hereafter referred to as Target Gas 5, was measured during 11 randomly chosen days, 20 times over that period, about 20 minutes each time."*

*In the revised version, Table 1 (now Table 3) and Figure 1 (now Figure 5) are presented in two different section, Sect. 3.1.2 and 3.1.4, respectively.*

**18.     line 210, can you add an uncertainty to the 2.2 ppb?**

The uncertainty will be added: $2.2 \pm 0.1$ ppb.

*Lines 270-271 "The same gas was also measured by GC-FID coupled to a preconcentrator, yielding a $C_2H_6$ mixing ratio equals $2.2 \pm 0.1$ ppb."*

**19.     line 212, this was a surprisingly high offset. I am also still getting confused by the working gases used. This is apparently not the same one used for Figure 1? And if Figure 1 averaged 23 ppb, presumably you were giving it less than 2.2 ppb (2.2 * 23/33 = _1.5 ppb?).**

As explained in question 17, different working gases were used for time drift (Figure 1) and Linearity test and CMR and Allan Deviation (Table 1 and Table 2). The nominal value for working gas of measured 23 ppb was unknown (question 17). However, as it was filled with ambient air, it has a $C_2H_6$ concentration similar to another working gas, so possibly, 2.2 * 23/33 = ~1.5 ppb.

*lines 272-277 "This value suggests a bias of the CRDS instrument of 31 ppb at low $C_2H_6$ concentrations, which is on the level observed for the ambient air. This bias comes probably from the fact that Target Gas 2 concentration is not known with a precision good enough, leading to errors when diluting to very low concentrations. To remove this bias, $C_2H_6$ mixing ratio were taken as enhancements over background during mobile measurements (Sect. 3.2 and 3.3). For more demanding purpose, a calibration strategy with more measurement points in the lower $C_2H_6$ concentration range and calibration tanks with lower uncertainty should be used."*

**20.    line 218 and 220, there are a lot of Picarro model numbers in this paragraph. Perhaps the authors could add a table to show what models measure what species, since I am not familiar with all the models.**
The table with instrument characteristic will be added, in the methods section, in paragraph 2.1 Laboratory set-up, before comparing different instruments.

*Lines 111-113 "Results obtained for CRDS G2201-i are compared with performances of CRDS G2132-i, which also can measure C2H6 as additional feature (Rella et al. 2015) and CRDS G2210-i, which is designed for C2H6 measurements. The characteristic of each instrument is presented in Table 1."*
*Line 115 Table 1 Characteristics of the instruments used during the study*

**21.    Figure 2, what units are the Allan deviation plots in? I assume they are all ppm?**
Yes, on Figure 2 Allan Deviation is presented in ppm. The axis labels will be improved in revised version.

*In the revised version Allan deviation is presented on Figure 4 (line 293).*

**22.    line 240, in some cases such as these, a standard error of the mean would also be worth reporting, along with the standard deviation**
The standard error will be calculated for all three protocols and their values will be added in the text. The difference between standard deviation and standard error will be also explained in the text.

*lines 249-250 "Applying the low humidity correction values, the $C_2H_6$ average value is $28 \pm 61$ ppb (standard error 22 ppb), which is similar to the $C_2H_6$ average value obtained during CMR test ($33 \pm 51$ ppb for raw data), in dry air (Sect. 3.1.3)."*

**23.** *Figure 3, are the differences between Protocol 2 and 3 simply linear fits, i.e., Protocol 2 fits a line to all the data, and Protocol 3 fits a line to data < 0.16% $H_2O$? If so, what would a higher-order fit to $H_2O$ do – could you use that for both high and low humidity cases? I'm also not sure of the benefit of naming these "Protocol X", since every time they are mentioned, a description of the Protocol is also given. It seems easier to mention "no correction", etc. every time, and the reader wouldn't have to remember what arbitrary Protocol number this was given.*

All 3 protocols fit all the data but protocol 1 uses no correction, protocol 2 uses the high-water content equation on all data (except the first point at 0%) and protocol 3 uses the low-water content equation on all data. The name "protocol X" will be deleted from text and it will stay with "no correction", "low humidity", "high humidity" as suggested by the reviewer.

*lines 128-130 "According to that study, if the water vapor level in the measured gas is less than 0.16% ("low humidity case"), then interference correction factors are the same for both devices. In the presence of water vapor (=>0.16 %, "high humidity case"), the correction factors were different for each device."*

*lines 141-143 "During data analysis, the interference correction factors from Assan et al. (2017) were applied (Table 2). Three cases were tested: no correction, high humidity case and low humidity case (except for the first step with dry air, where only the low humidity correction was applied)."*

**24.** *line 255, what does a release with a $C_2H_6$:$CH_4$ equal to 0 mean? No ethane was released, but methane was? Or nothing at all was released?*

In the case when $C_2H_6$:$CH_4$ was equal to 0.0, yes, ethane was not released while methane was released. This information will be added in the manuscript.

*lines 319 "In the case, when $C_2H_6$:$CH_4$ = 0.00, ethane was not released while methane was released."*

**25.** *line 264, when absolute deviations are on the order of 10 ppb, an "improvement" of 0.4 ppb seems like simple statistical variation. In other words, I think the authors are assigning significance to the insignificant digits of these numbers.*

We agree with this comment and this part will be rewritten to show insignificant change of observed variation between raw and averaged data.

*This sentence was removed in the revised version.*

**26.** *Table 3, Why do the authors report the residuals, and not the ratio itself? And how are the residuals defined? Is a linear fit performed on the data, and these are the residuals when the fitted line is subtracted from the data?*

Here, residuals are presented instead of the ratio to present the difference between emitted and observed ratios. In the revised version, the table will be improved to present ratios themselves.

Yes, to obtain these residuals, the fitted line is subtracted from the data. This part will be clarified in the revised manuscript.

*Table 3 (in the revised version Table 5) is improved. In the revised version, 10 s averaged data are deleted and only non-averaged data are kept. Now, $C_2H_6$:$CH_4$ are presented and their residuals, both for standing inside the plume situation and AirCore sampling.*

**27.     Table 4, referring to the different sites as A, B, C, and D only further complicates this table. Also, I'm not yet sure what difference the survey number makes. I think it would be easier to refer to these as compressor 1, 2, 3, and landfill. Use abbreviations if necessary. Also, move the * information from the title of the table to below the table.**

In Tables 4 and 5, numbers represent different measurements made on one site (e.g. made during different days or in different location on the site). In the revised manuscript, the terminology of sites will be clarified and the * will be moved below table.

*In the revised version sites are named Ga, Gb, Gc and L. The number of surveys are kept (Table 6, Table 7, Table D1, Figure 7 and Figure 8).*

**28.     Figure 5, how are these slopes calculated? Are the data weighted in the fit? And are the uncertainties reported in Table 4 just the slope uncertainties, or do they tie in the uncertainties of the C2H6 measurement?**

As described in question 12, slopes are calculated using a linear regression type II (uncertainty of x- and y-axis influence fitting) with the ordinary least squares (OLS) method. The data are not weighted. Reported in Table 4 and Table 5 uncertainties are slope uncertainties without adding uncertainties of $C_2H_6$ measurements.

*lines 359-362 "$C_2H_6$ and $CH_4$ mixing ratios are taken as enhancements over background. Slopes are calculated using a linear regression type II (uncertainty of x- and y-axis influence fitting) with the ordinary least squares (OLS) method. The data are not weighted. Uncertainties reported in Table 6 and Table 7 are linear fitting slope uncertainties without adding uncertainties of $C_2H_6$ measurements."*

**29.     line 407, it might be best to reiterate the requirement that $CH_4$ be greater than 1 ppm here, as mentioned previously in the paper**

This part will be rewritten to highlight possibilities of using CRDS G2201-i to measure ethane to methane ratio and the requirement of 1 ppm of $CH_4$ will be added there.

*lines 486-488 "Also, this instrument can be used to observe possibly temporal variation of $C_2H_6$:$CH_4$ of methane emitted from fossil fuel sources. These studies can be made in the vicinity of strong emitting sources, where $CH_4$ plume reaches at least 1 ppm above background."*

**30.     Grammar suggestions/typos:**

*line 28, it looks like "sources" is possessive, needs apostrophe*

*line 34 and elsewhere, I think "ethane:methane ratio" is redundant. Suggest either "ethane:methane" or "the ethane to methane ratio". But to me, using a colon implies ratio.*

*line 42–43, change "methane enhancement source" to "methane source"*

*line 45, remove "access to"*

*line 56, change "biogenic or thermogenic" to "biogenic from thermogenic"*

*line 85, add "1" to H in CH4 for consistency*

*line 124, change to "Equation"*

*line 133, change "has been measured during" to "was sampled for"*

*line 152, change "the previous works" to "previous work"*

*line 155, change "standing some" to "spending"*

*line 155, change "accumulating air in" to "sampling air using"*

*line 159, add "the" before "C2H6:CH4"*

*lines 164–5, start sentence "A description of the experimental: : :", replace "find" with "found", and add period after "(2017)"*

*line 167, suggest "up to" instead of "until"*

*line 169, suggest "C2H6:CH4" instead of "ethane:methane" for consistency*

*line 170, suggest "stationed in the plume"*

*line 171–172, suggest ": : : the time spent within the plume was approximately 15 to 20 minutes."*

*line 173, suggest "tracer release"*

*line 174, change "5 liters"' to "5-liter". Also line 284.*

*line 175, change to ": : : bags were sampled inside : : : and one was sampled : : :"*

*line 175, change "bags" to "bag samples"*

*line 180, delete "real". I think "field" is sufficient.*

*line 183, add "the" before "C2H6:CH4"*

*lines 185–186, change "part of measurements without dryer" to either "part with a dryer" or "part of the measurements without a dryer"*

*Figure 1 caption, I would re-word and make two sentences, change "20 minutes" to "20-minute", start new sentence with "For each measurement point, squares represent: : :"*

*line 214, change to "As a result: : :"*

*line 218, change "dedicated to the measure of ethane" to "designed to measure ethane"*

*line 227, change "ethane absolute value" to "an absolute value of ethane"*

*line 229, change "deduct" to "deduce"*

*line 259 and elsewhere, suggest "stationary in-plume situation" instead of "plume standing situation"*

*line 273, suggest something like: "For the higher emission, the measurements and results were combined when the emission rates were 70, 72, and 73 L/min."*

*line 276, add "the" before "AirCore"*

*line 285–286, add "to" after "equal"*

*Table 4, change "Data" to "Date"*

*line 321, change "due the very" to "due to the very"*

*line 329, change "ratio" to "ratios"*

*line 356–357, the time of sampling is confusing. The first sentence makes it sound like the instrument spends 10 minutes online, followed by 10 minutes offline. The next sentence makes it sound like the instrument spends 10 minutes online, followed by 20 minutes offline.*

*line 358, perhaps just describe the CRDS data as being averaged over the sampling time of the GC-FID*

*line 360–361, change to ": : : to use a CRDS G2201-i to measure C2H6:CH4, : : :"*

*line 366, change to ": : : on the TILDAS method : : :"*

*line 367, change to "tracer release"*

*line 384–385, a word is missing here, perhaps "allowed us to", change "measurements point" to "measurement points"*

*line 391, change to either "allow us to separate" or "allow the separation"*

*line 394, change to "flask samples"*

*line 398, indicative of what?*

All suggested grammar correction and found typos by Reviewer will be corrected and after the revised manuscript will be verified again with a view to the grammar and typos. Also, the sentences in lines 356-358 will be rewritten to make it clear and consistent. The sentence in line 356 will be rewritten: "For GC-FID, ambient air was collected 10 minutes and during following 20 minutes instrument measured the input air."

*In the revised version all suggested grammar corrections are implemented and typos are corrected.*

*for the tables in the Appendix, I would put the \* asides below the table, rather than part of the table title*

The * will be moved below table.

*The * is moved below the table.*

We would like to thank Reviewer 3 for the constructive comments that aided us to improve our manuscript. In this document we provide our replies to the Reviewer's comments. The original comments made by the Reviewer are typeset in italic and bold face font. Following every comment, we give our reply. Here line numbers, page numbers and figure numbers refer to the original version of the manuscript. The authors' changes in the manuscript are typeset in italic face font and line number refer to the new version of the manuscript.

*General Comments*
*This manuscript assesses the ethane measurement obtained using the Picarro G2201- i and tests its ability to provide meaningful data for determining C2H6:CH4 in methane plumes, with the goal of source attribution. The instrument is tested and calibrated in the laboratory, subjected to controlled release experiments, and taken to measure real sources in the field. The authors find that, due mostly to the low precision of the ethane measurement (~ 50 ppb), the G2201-i can only realistically be used for ethane-to-methane ratios in methane peaks that are at least 1 ppm above the background. Furthermore, the measurement as presented must be taken under stationary conditions (i.e., with the mobile platform parked within a plume for ~30 min) or the noise of the ethane measurement becomes unacceptably high.*
*The use of the G2201-i for the described applications seems extremely limited, especially in light of the other available instruments that can do this type of measurement much better (LGR, Aerodyne, and other Picarro models). However, the authors do recognize that in order to use the Picarro G2201-i for ethane field measurements (which in turn are to be used only in the calculation of ethane-to-methane ratios rather than absolute ethane mixing ratios), the instrument response must be extensively characterized. This work is done, and the limitations of the G2201-i for the purposes described are appropriately determined and discussed.*

*1.        There is a lot of information presented on the experimental details of previous work, which, in my opinion, obscures the experimental design and the main conclusions of the current manuscript somewhat. It makes it difficult for the reader to focus on the important points of the manuscript (one of which is the many conditions that need to be satisfied to obtain useful ethane information from the G2201-i). I recommend the authors try to streamline the manuscript as much as possible so that the important points are evident. Additionally, I recommend careful proofreading of the manuscript, which contains many small grammar errors, some of which are highlighted below under "Technical Comments".*
The manuscript will be "streamlined" as suggested to present the study more clearly in order to be useful for other scientific teams which already have CRDS G2201-i and would like to use it in field conditions for measuring both $\delta^{13}CH_4$ and ethane to methane ratio. Thus, we will modify our manuscript for that. It can be treated as a protocol where all necessary steps are described and verified before field work. To do it, in the revised manuscript the introduction and conclusion parts will be improved to highlight the importance of the work done. Also, the method section will be rewritten to make it more straightforward and some additional explanation will be added to make it more "stand-alone" work (e.g adding scheme or conducted test before using instrument on field or adding table to equation 1, with factors A, B, C for different humidity levels).

*lines 76-86 "Here, the main purpose of this study is to evaluate the performances of the CRDS G2201-i and the applicability of making short-term, direct, continuous, mobile measurements of ethane in methane-enriched air, with sufficient precision during near-source surveys. Our motivation is to perform both isotopic and ethane measurements with only one instrument in the field in order to improve the partition of methane sources without the need for an additional analyzer. We aim at providing a protocol useful for other scientific teams, that do not have an analyzer designed for ethane measurements, but already have the CRDS G2201-i and intend to use it in field conditions for measuring both $\delta^{13}CH_4$ and ethane to methane ratio.*

*To achieve this goal, the first step consists of laboratory tests to calculate the calibration factors and also to check the instrument performances in stationary, laboratory conditions extending preliminary work by Assan et al. (2017). The second, novel step evaluates the performances of the instrument during mobile field measurements in a controlled release experiment. A mixture with known $C_2H_6$:$CH_4$ and $CH_4$ emission flux was released and compared to measured ratios from CRDS G2201-i and LGR UMEA."*

*lines 106-107 "Figure 1 Flow chart of steps to use $C_2H_6$ measured by CRDS G2201-i. The number in the corner corresponds to the subsection where methods of each step are presented."*

*lines 118-124 "The cross sensitivities with $H_2O$, $CO_2$ and $^{12}CH_4$ induce a bias in raw $C_2H_6$ observed by CRDS G2201-i. Assan et al. (2017) provided the strategy to determine $C_2H_6$ correction factors to account for these interferences. During the experiment, the $C_2H_6$ mixing ratio of measured gas mixture was constant, while the mixing ratio of interfering species was changed and controlled using a setup similar to the one presented on Fig. 2 in the Sect. 2.1.2. During one measurement set, the concentration of only one interfering species was changing, while the concentration of other species stayed stable. The measurement set was repeated while varying concentrations of $H_2O$, $CH_4$ and $CO_2$ were adjusted. Using linear regression, the test yielded values for the interference correction factors A, B, C in Eq. 1:"*

*line 135 Table 2 Interference correction on $C_2H_6$ (Assan et al., 2017)*

*line 164 Figure 2. Experimental setup used during laboratory tests.*

*lines 351-353 "During the release experiment, we showed that the CRDS is able to separate the different emitted mix through their $C_2H_6$:$CH_4$. Standing in the plume resulted in a better agreement with the real ratios, with less spread of the residuals than using AirCore sampling. Increasing the AirCore sampling frequency could potentially help resolve this discrepancy."*

*lines 406-408 "Field work allowed us to compare our measurements against operator values and GC measurements. This confirms that this instrument can discriminate between sources and that it agrees within its uncertainty with more precise methods such as GC."*

*lines 444-446 "Even though, we showed it is possible to receive reliable values during short time (e.g. 35 minutes) and the instrument can be installed inside a car. Notably, having the instrument setup inside the car facilitates the measurement setup, as an additional place to install the stationary instrument is not required anymore."*

*lines 486-488 "Also, this instrument can be used to observe possibly temporal variation of $C_2H_6$:$CH_4$ of methane emitted from fossil fuel sources. These studies can be made in the vicinity of strong emitting sources, where $CH_4$ plume reaches at least 1 ppm above background."*

**Specific Comments**
**2.      Lines 53-58: How does this study differ from Assan et al? Is the system just characterized better? Is the only difference, as mentioned later in lines 361+, that the instrument was put in a car (which must remain stationary within a plume for ~30 mins to take a useful measurement)? If so, that should be made clear early on.**
In our study, additional tests were made, the previously calculated correction and calibration factors were evaluated and long term drift was verified. Notably, we did not observe the time drift, contrary to Assan et al. Also, compared to Assan et al., a controlled release experiment was made. Ultimately, we wanted to check in which conditions we can measure ethane to methane ratio in short time, near-source conditions. Overall, we showed it is possible to received reliable values during short time (i.g. 30 minutes) and the instrument can be installed inside the car. Having the instrument set-up inside the car facilitates the measurement set-up as an additional place to install the stationary instrument is not required anymore. However yes the measurements are field-tested with the car idling. This explanation will be added in the method section.

*lines 82-84 "To achieve this goal, the first step consists of laboratory tests to calculate the calibration factors and also to check the instrument performances in stationary, laboratory conditions extending preliminary work by Assan et al. (2017). The second, novel step evaluates the performances of the instrument during mobile field measurements in a controlled release experiment."*
*lines 444-446 „Even though, we showed it is possible to receive reliable values during short time (e.g. 35 minutes) and the instrument can be installed inside a car. Notably, having the instrument setup inside the car facilitates the measurement setup, as an additional place to install the stationary instrument is not required anymore."*

**3.      Lines 62+: Did you use the monitoring mode in addition to the replay mode for the Aircore in the current study? I think some more information on how the Aircore was used specifically for this study should be included, although I would add this information later in the methods section.**
Indeed, all measurements which were made during standing inside the plume were made in the monitoring mode. Also, we drived through $CH_4$ plumes (in monitoring mode) and remeasured air accumulated in the AirCore sampler (replay mode). However, during car motion, the instrument noise increased and also crossing road bumps can cause additional fluctuation of measured $C_2H_6$. Thus, comparison of data collected in monitoring mode and replay mode, where the same plume is remeasured, can by biased due to influence of car motion for $C_2H_6$ readout in monitoring mode. This information will be added in the method section.

*lines 195-202 "Here, both stopping inside the plume and AirCore approaches were used during mobile measurements. The AirCore used in this study is made of 50 m Decabon storage tube. In our setup, the instrument flow rate in the monitoring mode was increased to 160 mL min$^{-1}$ (by default, in CRDS G2201-i the flow rate is equal to 25 mL min$^{-1}$) to achieve faster instrument*

*response during mobile measurements. Then, the flow rate in the reply mode was chosen as the optimal solution between increasing the number of measurement points and having enough air for each zone sampled. Here, in the replay mode, using the needle valves, the flow rate decreased about 3 times. With 50 mL min⁻¹ flow rate, one AirCore analysis lasts about ten minutes. In the replay mode, the car was stopped to avoid possible increase of instrumental noise due to car vibration. While stopping inside the plume, the data were collected in the monitoring mode with engine stopped."*

**4.      Lines 82 – 85: This background information on how ethane is measured and reported for an isotopic methane/carbon dioxide instrument should be moved to the abstract and introduction.**
This information will be moved to the introduction section.

*lines 60-63 "Here, building on previous studies with CRDS instruments, we detail the possibilities and limitations of measuring $C_2H_6$ using the CRDS G2201-i, in the vicinity of methane source. The CRDS G2201-i is originally designed to measure $^{12}CO_2$, $^{13}CO_2$, $^{12}CH_4$, $^{13}CH_4$ and $H_2O$ and record $C_2H_6$ only as an internal way to correct $^{13}CH_4$, thus observed $C_2H_6$ mixing ratio must be corrected and calibrated."*

**5.      Lines 81+ (Materials and Methods section): To make each factor investigated clear, consider re-formatting with subheadings, such as, 1.1 Laboratory 1.1.1 Interference Correction on Ethane and Water Sensitivity 1.1.2 Ethane Calibration Factors 1.1.3 Precision and Allan Variance 1.1.4 Time drift Because the water vapor sensitivity tests are tests on the validity of the interference corrections, I think this should be discussed at the same time as the interference correction in general.**
Material and Method section will be rewritten according to this comment. These changes will be followed by changed order in the results paragraph.

*In the revised manuscript, for the laboratory tests, the order was changed as follows, both for Material and Methods section and Result section:*
*2.1. Laboratory setup*
*    2.1.1. Sensitivity of interference correction parameters to humidity*
*    2.1.2. Ethane Calibration Factors*
*    2.1.3. Precision and Allan Variance*
*    2.1.4 Time drift*

**6.      Lines 147+: I have some confusion about what Protocol 1, 2, and 3 are. Are these described clearly somewhere? I would add relevant details here in the methods section.**
Protocols 1,2,3 are arbitrary made protocols to describe which correction factors were used. Thus Protocol 1 is when no interference correction was used, Protocol 2 when interference correction was used for high humidity case and Protocol 3 for low humidity case. However, as Protocol X is always followed by a short description of cases, in the revised manuscript we will remove "Protocol X" and will mention for which case results are presented (i.e. "no correction", "high humidity" or "low humidity").

*lines 128-130 "According to that study, if the water vapor level in the measured gas is less than 0.16% ("low humidity case"), then interference correction factors are the same for both devices. In the presence of water vapor (=>0.16 %, "high humidity case"), the correction factors were different for each device."*

*lines 141-143 "During data analysis, the interference correction factors from Assan et al. (2017) were applied (Table 2). Three cases were tested: no correction, high humidity case and low humidity case (except for the first step with dry air, where only the low humidity correction was applied)."*

**7.      Lines 151-152: Delete "The measurement setup used here is the same as in the field" and only mention in section 2.3.**
This correction will be applied.

*In the revised version, the general description of mobile measurement is presented in Sect. 2.2 Mobile measurement setup. Then, the detailed setup for controlled release experiment is presented in Sect. 2.3 and for field setup in Sect. 2.4.*

**8.      Lines 154-159: The point that true "mobile" measurements are not conducted (i.e., while the vehicle is moving) should be highlighted earlier in the manuscript. It is an important point that is somewhat hidden here. Also- please add information here about the specifics of the Aircore setup as used in this study (e.g., flow rates, different modes, car stopped or moving).**
This information will be added in the method section (question 3). Also the details about how to use AirCores will be added.

*lines 195-202 "Here, both stopping inside the plume and AirCore approaches were used during mobile measurements. The AirCore used in this study is made of 50 m Decabon storage tube. In our setup, the instrument flow rate in the monitoring mode was increased to 160 mL min$^{-1}$ (by default, in CRDS G2201-i the flow rate is equal to 25 mL min$^{-1}$) to achieve faster instrument response during mobile measurements. Then, the flow rate in the reply mode was chosen as the optimal solution between increasing the number of measurement points and having enough air for each zone sampled. Here, in the replay mode, using the needle valves, the flow rate decreased about 3 times. With 50 mL min$^{-1}$ flow rate, one AirCore analysis lasts about ten minutes. In the replay mode, the car was stopped to avoid possible increase of instrumental noise due to car vibration. While stopping inside the plume, the data were collected in the monitoring mode with engine stopped."*

**9.      Line 174- 178: I question whether any of the information about the failed bag measurements should be included in the main manuscript, especially given the issues with sampling and bag preparation mentioned later (in lines 284+). Maybe make a very abbreviated reference to them, and then move all other bag information to the supporting information.**
During preparing the manuscript, we considered the same question. In the revised manuscript, we will move this bag samples part to Appendix and leave only a short explanation in the main text.

*lines 222-226 "Three other releases were measured using sampling 5-liter bags (Flexfoil, SKC Inc.) only. Between 1 and 3 bag samples were sampled inside the plume and one was sampled outside as a background sample. Afterward, bags samples were measured in the laboratory using the CRDS G2201-i. The samples were measured without drying and the correction was applied for water vapor higher than 0.16 % ("high humidity case"). Then the $C_2H_6$:$CH_4$ enhancement ratio was calculated for every bag separately and also as a regression slope of $C_2H_6$ against $CH_4$ values. Results are presented in Appendix C."*

**10.     Lines 195+, Section 3.1: Suggest headings that are the same as those suggested above for section 2.1 to help organize the information.**
The heading will be changed and text will be rewritten according to suggestion in question 5.

*In the revised manuscript, for results section, the headings are changed as for material and methods section (question 5).*

**11.     Line 201: Can you specify what a "low amount of C2H6" means?**
This working gas was fulfilled with dried ambient air, thus $C_2H_6$ concentration was similar to concentration of 2.2 ppb in working gas used to measurements of CMR and Allan Deviation. In the revised manuscript, used working gases will be better numbered and better described.

*lines 180-182 "A known working gas (dry atmospheric mixing ratio of $CH_4$ and $C_2H_6$), hereafter referred to as Target Gas 5, was measured during 11 randomly chosen days, 20 times over that period, about 20 minutes each time.*

**12.     Line 356: Please clarify what "10 mins of ambient air collection was measured during 20 minutes" means.**
Sentences in lines 356-358 will be rewritten to make it clean and consistent. The sentence in line 356 will be rewritten: "For GC-FID, ambient air was collected 10 minutes and during following 20 minutes the instrument analysed the input air."

*lines 432-433 "To have identical timestamps as GC-FID, corrected and calibrated CRDS data were averaged for 10 min every 30 min."*

**13.     Lines 384-390: Please revise this section on Aircore and CRDS flow rates for clarity. How are the Aircore and CRDS flows related? Were there reasons for the chosen flows?**
In the set-up used in this study, the CRDS had flow 160 mL/min and this flow rate was used in the monitoring mode. Then in the replay mode, using the needle valves, the flow rate decreased to about 50 mL/min. It caused the three times increase of the number of measurement points. In our set-up, the instrument flow rate in monitoring mode was increased (by default, in CRDS G2201-i the flow rate is equal to 25 mL/min) to achieve faster instrument response during mobile measurements. Then, the flow rate in the reply mode was chosen as optimal solution between increasing the number of measurement points and having enough air for each zone sampled. With 50 mL/min flow rate, one AirCore analysis lasts about ten minutes. Smaller flow rate would allow to increase the number of measurement points and thus instrument precision, but also requires longer time to use replay mode and measure air stored in AirCore. The AirCore tool was also used

inside Paris city and we decided to keep 10 minutes sampling time of replay mode. As Paris is a crowded city with numerous traffics, it would not be possible to stop the car for longer analysing time to measure $CH_4$ plume in the replay mode. The additional explanation of AirCore and CRDS flow rates will be added.

*In the revised manuscript the detailed description of used AirCore is added in Sect. 2.2 (questions 3 and 8).*

**14.** **Line 398: Do you mean the "first comparisons" of ethane mixing ratios with GC-FID match up in a relative sense? The word "indicative" is confusing here.**
In this line we mean first comparison between CRDS sampling according to the protocol presented in the manuscript with GC-FID. The sentence will be rewritten for clarity.

*lines 479-481 "Nevertheless, due to the short sampling time of the flasks, these first comparisons between flask samples measured by GC-FID and short-term CRDS field measurements are only approximate and more comparison campaigns should help to understand the discrepancies between these instruments."*

**15.** **Line 431: Please clarify what the "flushing issue" to be solved is.**
Here, by flushing issue we mean a too small decrease of the flow rate. A stronger decrease of the flow rate will result in an increase of measurements points and an improvement of instrument accuracy. Further decreasing flow rate could solve the problem of observed differences between AirCore and actual ethane to methane ratio (discussion in lines 382- 390). In line 431 it will be rephrased for clarity.

*lines 514-515 "To fix this problem, $C_2H_6$:$CH_4$ can be measured by standing inside the plumes or offline using AirCore sampling after determining the optimal flushing flow (see Sect. 2.2 and 3.2)."*

**16.** **Technical Comments**
**Line 43: "source" should be "sources"**
**Line 54: "measure of" should be "measurement of"**
**Line 60: Change "allows to improve time resolution" to "allows improvement of time resolution"**
**Line 77: Change "instrument to ethane" to "instrument for ethane"**
**Line 165: Change "find" to "found"**
**Line 167: Change "emission flux" to "gas flow rate"**
**Line 168: Change "could vary" to "were varied"**
All suggested grammar correction and found typos by Reviewer will be corrected and after the revised manuscript will be verified again with a view to the grammar and typos.

*In the revised version all suggested grammar corrections are implemented and typos are corrected.*